# Towards the Idea of Molecular Brains

**DOI:** 10.3390/ijms222111868

**Published:** 2021-11-01

**Authors:** Youri Timsit, Sergeant-Perthuis Grégoire

**Affiliations:** 1Aix Marseille Université, Université de Toulon, CNRS, IRD, MIO UM110, 13288 Marseille, France; 2Research Federation for the Study of Global Ocean Systems Ecology and Evolution, FR2022/Tara GOSEE, 3 rue Michel-Ange, 75016 Paris, France; 3Institut de Mathématiques de Jussieu—Paris Rive Gauche (IMJ-PRG), UMR 7586, CNRS-Université Paris Diderot, 75013 Paris, France; gregoireserper@gmail.com

**Keywords:** networks, signalling, behaviours, information processing, ribosome, nervous systems, brains, allostery, aromaticity

## Abstract

How can single cells without nervous systems perform complex behaviours such as habituation, associative learning and decision making, which are considered the hallmark of animals with a brain? Are there molecular systems that underlie cognitive properties equivalent to those of the brain? This review follows the development of the idea of molecular brains from Darwin’s “root brain hypothesis”, through bacterial chemotaxis, to the recent discovery of neuron-like r-protein networks in the ribosome. By combining a structural biology view with a Bayesian brain approach, this review explores the evolutionary labyrinth of information processing systems across scales. Ribosomal protein networks open a window into what were probably the earliest signalling systems to emerge before the radiation of the three kingdoms. While ribosomal networks are characterised by long-lasting interactions between their protein nodes, cell signalling networks are essentially based on transient interactions. As a corollary, while signals propagated in persistent networks may be ephemeral, networks whose interactions are transient constrain signals diffusing into the cytoplasm to be durable in time, such as post-translational modifications of proteins or second messenger synthesis. The duration and nature of the signals, in turn, implies different mechanisms for the integration of multiple signals and decision making. Evolution then reinvented networks with persistent interactions with the development of nervous systems in metazoans. Ribosomal protein networks and simple nervous systems display architectural and functional analogies whose comparison could suggest scale invariance in information processing. At the molecular level, the significant complexification of eukaryotic ribosomal protein networks is associated with a burst in the acquisition of new conserved aromatic amino acids. Knowing that aromatic residues play a critical role in allosteric receptors and channels, this observation suggests a general role of π systems and their interactions with charged amino acids in multiple signal integration and information processing. We think that these findings may provide the molecular basis for designing future computers with organic processors.


*Undoubtedly, only artists devote themselves to science…*

*(Santiago Ramon y Cajal)*


## 1. Introduction

At about the same time that Aristotle (384–322 BC) was elaborating perhaps the first organised reflections on “what is life?” in his work “*Peri psyches*—On the soul” [1], Zhuangzi (389–319 BC) was dreaming that he was a butterfly [2,3]. However, when he woke up, he wondered if it was not the butterfly that was dreaming of Zhuangzi? This dream was so lively that one still wonders if Zhuangzi really existed. However, the butterfly’s dream also transports us into a question that complements that of Aristotle: “what is reality, and how does human, butterflies and more generally, life sense it, build it and navigate between themselves, auto-stimulation and their perception of the universe. Intriguingly, these Western and Eastern thoughts also come together in the following strange convergence: in ancient Greek, *psyches*, “the soul”, also means “butterfly”.

While “this is a butterfly!” is the most common response on cards 1 and 5 of the Rorschach projective test, whose ink stains can reveal the deep structure of the human mind [4,5,6], monarch butterflies are also a gateway to quantum mechanics in biology. Thanks to their cryptochromes, which contain pairs of radicals, monarch butterflies perceive the Earth’s magnetic field and their wingbeats, can guide them over thousands of kilometres and bring us into the fascinating world of quantum biology [7,8,9]. The flapping of their wings that has also become a famous metaphor for chaos [10] could equally well illustrate the processes of cell signalling amplification, in which the weak stimulation of a receptor by a ligand or a photon, *the flap of a butterfly’s wings in Brazil*, can set off a cascade of molecular events that induce the overall change in the behaviour of an organism, *a tornado in Texas*. Thus, “butterfly effects” illustrate unexplored lands that touch both the essence of life and soul. First, they translate in a poetic way the evanescence of one of the fundamental properties of life, which still escapes the investigation of modern science: what animates it and does it have a will? This notion that Aristotle raised very early on by attributing a *psychès*, meaning a “soul” or “a form” not distinct from the body, to all living beings, be they plants or animals, is surprisingly topical in the light of a set of recent works on “*consciousness*” in single cells and organisms without a nervous system [11]. Second, the butterfly’s dream addresses the question of doubt and incertitude principles, where philosophy and quantum mechanic could meet in their fundamental question about our relationship with the reality and the limits of measurments to apprehend it. Let us remember that much later in 1641, Descartes in his first Meditation will give us more or less the same dream as Zhuangzi and concluded: “All that I have received so far for the truest and most assured, I have learned from the senses, or through the senses: but I have sometimes felt that these senses were deceptive, and it is prudent never to rely entirely on those who have once deceived us” [12]. Finally, butterflies illustrate the volatile and moving world, woven of order and disorder where life likes to lodge itself for playing hide and seek with simplistic mechanistic principles.

Speaking about “brains”, even molecular ones, and “consciousness”, even cellular ones, are intended to examine what comes from oneself and what comes to us from the outside. Such precautions regarding reality are not only metaphysical and concern, as closely as possible, the nature of life and how its components perceive and respond to external or internal fluctuations, from the nano- to the macroscopic scales. In his seminal paper of 1995, Dennis Bray proposed that proteins can form circuits equivalent to the nervous system within cells [13]. However, he also suggested an equivalence between the “intrinsic activity” of the brains (as for example dreaming) and of the cells [14]: “both higher mammals and single cells (and by implication everything in between) generate movements in a proactive mode”. Supported by many experimental observations, this view considers that cells or organisms “continually rehearses possible future actions” that are “selected from an upwelling of spontaneous activity that serves to anticipate incoming stimuli” [14] (and references cited in). Can unicellular organisms, organelles and even macromolecular complexes such as ribosomes or stressosomes dream, be lured or confuse internal and external stimuli? According to a recent review by Mitchell and Lim, the answer would be yes: and cells would be prone to misperceptions, analogous to visual illusions, sometimes leading to incorrectly decoding input patterns of their environment [15]. Interestingly, these observations can also be extended to the molecular level. It is now admitted that, rather than being simple switches (ON/OFF), membrane receptors that pre-exist in a balance between active and inactive conformation can be activated spontaneously in the absence of a ligand [16,17]. More than a simple phenomenon of scale, this transposition could be the occasion to reconsider under another angle the biological macromolecules that “already” behave as complex systems [17,18,19,20,21,22]. Could the “emergence” phenomena from molecular and neural networks provide a kind of “scientific form” to the Aristotle *psychès*? Discussing “Molecular vitalism” 20 years ago [23] or “nano-intentionality” a little later [24] already incited to think differently about life and firstly questioned the pertinence of its hegemonic “machine metaphor”. In an odd paradox, Descartes, in spite of his awareness of the relativity of the interpretation of senses, is also at the origin of the “machine metaphor” in biology. Many years after his “meditations”, he proposed the analogy between human, animals or their organs with human-made machines that have imposed in a lasting way the mechanistic conceptualisation of life until today. However, this anthropomorphic conception of life starts to be seriously challenged, and it could be asked if the machine metaphor may hinder access to the essence and complexity of life.

### 1.1. Feeding Without Feedback: Chaplin and the “Machine Metaphor” of Life

In modern times of biology, the “machine metaphor” remains the pillar of the conception of life and its constituents. The ideas of “high performance” or “efficiency” of biological molecules optimised by selective forces are an integral part of the symbolic field of “a technological power”. In the collective imagination, machines are the fundamental actors of the industrial world, and they constitute the absolute references of reliability and efficiency of a productivist ideology. With “Pacific 231” and “Iron Foundry”, Arthur Honegger and Alexander Mosolov both composed symphonic pieces to the glory of machines, whether they were west or east of the Iron Curtain. As symbols of strength and power, for the societies in which science is produced (as by a machine), machines could have established their hegemony over the conceptualisation of life and its components. In his film “Modern Times”, Charlie Chaplin had grasped in all its depth the limits of this metaphor and the fate that the era would reserve for life and its “machine” metaphors. In a scene that is probably one of the most comic scenes in cinema, a machine is supposed to automatically feed a worker (The Tramp, C. Chaplin) so that he does not interrupt his work on the line. This infernal machine gets out of control and forcibly introduces all kinds of food into the mouth of the unfortunate “Tramp”. This tragi-comic scene shows the filmmaker’s ingenious intuition in the sciences of complexity: *feeding without a feedback* system is irremediably doomed to failure (Figure 1). Later, the film also illustrates in a masterly way how without the concertation of the mRNA and the three sites of the A, P and E tRNAs in the ribosome, the synthesis of proteins can very quickly become catastrophic (Figure 2). Premonitory, would he anticipate the fateful fate that modern times have reserved for biodiversity? *Feeding without feedback* is indeed the antithesis of system biology, and such conceptualisation of life may severely hinder the understanding of the fundamental properties of its components. Thus following a strong selective pressure in the evolutionary history of ideas, the biological macromolecules are still today inexorably associated with molecular machines. We may wonder if it is not time to end this analogy, which dates back to the 17th century and which may bias or hinder our understanding of some of their properties that do not fit into this conceptual framework?

One sometimes finds in scientific literature: “Dancing in the cloud” [25], choreography [26,27] and symphonies [28] of life: rebel against anthropomorphic stereotypes, moving and escaping the dangers of reductionism, the metaphors coming from the artistic world seem much more relevant to conceptualise life and leave the way open for new ideas to emerge. For example, the choreographic metaphor seems to us to be much more appropriate than the machine metaphor for understanding how the concerted movements of molecules contribute to cellular life and its motility. However, although this metaphor has much richer heuristic virtues than machines for conceptualising life, it is little used compared to the machine (166 pubmed entries for “molecular choreography” versus 16,999 for “molecular machine”). Like cells, choreographic creation is based on the close relationship between the writing of a score (genotype), its interpretation (phenotype) and sometimes or even often, improvisation (epigenetics). In fact, if a choreography is generally written, the conventions of writing and transmission can be as numerous as the choreographers [29,30,31,32]. These different “codes” of expression correspond surprisingly to the great diversity of what is called cell signalling (see below), but they converge towards a single result: a global harmony of movements common to all cells. Although the gestures and movements of each dancer are precisely written and planned in time and space, they must constantly scrutinise each other discreetly to synchronise. The score alone is not enough to make a successful ballet. The interpretation, listening and constant adjustment of each dancer play a major role in the coherence, meaning and aesthetics of a choreography. This notion is interesting when trying to understand the constant interactions between macromolecules that must constantly readjust in a fluctuating environment. It is fascinating, for example, to observe by molecular dynamics, the dance of two DNA double-helices closely intertwined, constantly looking at each other, attracting and repelling each other in turn in a kind of premise of love [33,34,35]: this scene is closer to a “pas de deux” than worm gear set. Yet, although machines and choreography are not very reconcilable, they are sometimes used to describe the same object. The ribosome, for example, may be an allosteric “molecular machine” [36] that can perform “the choreography of protein synthesis” [37]. Thus, see the molecules dance, contemplate the phenomena of emergence in the networks and their complex system behaviour’s [38] at all scales suggests that it may be time to escape from the machine metaphor in order to overcome the epistemological obstacles that may hinder the deep understanding of “what is life” [39].

### 1.2. Information Transfer, Processing and Behaviours at the Molecular Scale

Once freed from the conceptual stranglehold imposed by machine metaphors, today’s scientific context is conducive to re-examining both the “molecular vitalists” concepts and Denis Bray’s hypothesis [13]. On the one hand, many articles have recently highlighted the behavioural richness of microorganisms or plants lacking a nervous system [40] (see Section 2: “the psychic life of microorganisms”). They again raise the question of whether and how complex tasks can be performed without a nervous system. Cells contain a wide variety of receptors for virtually sensing all-physical and chemical stimuli (Section 3). These multiple signals are perceived, transmitted and integrated by the complex signalling networks to produce appropriate cellular “behavioural” responses. Cell signalling networks, which are now well characterised, are mostly based on the transient interactions of their components [38,41,42,43].

On the other hand, new types of molecular networks have been recently discovered in the ribosomes [44,45,46]. These “neuron-like” ribosomal protein (r-protein) networks display new properties compared to other “classical” biological networks [47]: they form mostly permanent connections through long disordered filaments and phylogenetically conserved tiny interfaces that have been compared to “molecular synapses”. These networks have evolved to optimise the interconnections between ribosome functional centres, thus presenting a “functional” analogy with simple sensorimotor networks [46] (Figure 2). They also probably use a new type of allosteric mechanism involving key aromatic amino acids. Conserved motifs formed by charged amino acids and π-systems are distributed along the whole networks and complexify during evolution. This suggested that charge transfer, propagation of electrostatic perturbation or even quantum phenomena may distribute signals throughout the network for synchronising ribosomal functional sites or even more complex tasks.

### 1.3. The Roots of the Molecular Brain Metaphor

Because of their analogy with sensorimotor networks, r-protein networks initially suggested a metaphor that gradually evolved from “neuron-like circuits” at a molecular scale [44,45] to the one presented here: “the molecular brains”. However, the idea of molecular “proto-brains” has been previously proposed in the context of bacterial chemotaxis by Stock and his collaborators in the early 2000s [48,49,50]. These proto-brains, which consist of clusters of bacterial chemotactic receptors, have been shown to control bacterial motility in response to attractants or repellants. Already 20 before the proto-brains, the idea of “molecular brains” was germinating with the articles of Adler and Koshland proposing an analogy between chemotaxis and neurobiology [51,52]. More than a century ago, however, Charles Darwin and his son Francis already indulged in a similar analogy by extending the concept of the brain to plant roots: “It is hardly an exaggeration to say that the tip of the radicle thus endowed, and having the power of directing the movements of the adjoining parts, acts like the brain of one of the lower animals; the brain being seated within the anterior end of the body; receiving impressions from the sense- organs, and directing the several movements” [53,54].

### 1.4. Brains Beyond Connectomes

However, the idea of “molecular brains” refers to the concept of the brain, and brains are still difficult to define… In a kind of “Unanswered Question” (Charles Ives, 1908), Vion-Dury and Mougin asked: “finally what is a brain?” in their paper “Neuroscience sans conscience n’est que ruine de l’âme” (a title that refers to François Rabelais’s novel “Pantagruel”). In their phenomenological approach, the authors conclude that modern neurosciences only give access to “fragments of experience, to blurred and perhaps false images of processes, to the distant shadows of the mind” [56]. Today, several ways of conceptualising the brain coexist more or less peacefully depending on whether one is in the camp of neurosciences [57,58], evolutionary biology [59,60], physiology, psychoanalysis [6], network science [61], physics [62,63], mathematics [64] or astrophysics [65]. “What is a brain?” would remain without a coherent answer for a long time to come, since the concept of the brain is so impalpable, and perhaps just as impalpable as the notion of the “*psychès*” and life.

When referring to neuronal networks and their connections, for example, synaptic, one usually uses the term connectome; this connectome, called anatomical connectome, is a structural description of the brain, and its connectivity properties can help exhibit central hubs of neurons for information integration, regions of highly interconnected neurons and important pathways of information [66]. Beyond anatomical connectome, networks can also be built from direct or indirect recording of neuronal activity by inferring statistical dependencies between the neurons (or collections of neurons); in this case, we refer to them as functional connectomes [61]. Several tools have been developed to study connectomes (centrality, rich club, small-worlds, graph similarities) [66,67]. We will, however, not detail them here. Behaviours can modify these structures of interactions, which lead to new ways to explore the brain-behaviour relation through structure-function relation.

The function of the brain is a global phenomenon that cannot be reduced to how its different areas function separately, and the relation structure/function would suffer from such reduction. In other words, wishing to find a relation between the function of an anatomical subpart of the brain and its local structure does not seem convincing, and one would rather like to understand how this subpart fits, with respect to its structure and function, in the whole brain. Until now, one would study the anatomical connectome as it is a global structure that one can have access to. However, the knowledge of the connectome is not sufficient for a global understanding of the function of the brain. For example, abnormalities of the brain connectome are known to be related to psychiatric disorders [68,69]; however, even though modifications of the connectome are observed, it would be difficult to interpret the disorder without an underlying model of how these dysfunctions appear [70,71]. The connectome is information that models have to take into account [72]; in other words, it is information that constrains the collection of relevant models so that they can, in turn, help understand the pathways for information integration.

Overall the connectome enables to exhibit interactions between relevant variables that come into play in the functioning of a “brain”; however, not all “brains” have a straightforward anatomical connectome [62]; for example, at the scale of the cell, there is no clear notion of an anatomical network of interactions between macromolecules and messengers that intervene in cell signalling although there is convincing evidence that this cell signalling is the physical support for information integration and decision making [73,74]. When looking for unified tools for describing these “brains”, one can turn to the more flexible notion of functional connectome can still be defined for these new “brains” as it relies on statistical inference; more generally, we propose, in the last section of this paper, the Bayesian brain as a unifying computational framework for what a “brain” is.

What is the nature of signals that flow through neural connectomes, and how do brains process this information? Two years after the storming of the Bastille, the publication in 1791 of “De viribus electricitatis in motu musculari commentarius” by Galvani was to bring about a lasting revolution in science, and in particular in neuroscience, by showing the role of electricity in nerve conduction and muscle contraction [75,76]. More than a century after the birth of electrophysiology, the development of increasingly sophisticated techniques has made it possible to analyse and understand the nature of nerve impulses in greater detail [77,78,79]. Today, all this knowledge forms the basis of computational or system neuroscience and current theories of neural coding. It is now well established that neurons communicate through a train of electric impulses (spikes) or action potentials [80]. The general class of models for describing how neurons generate spikes is called integrate-and-fire models, one of which is the Hodgkin-Huxley set of differential equations. Neurons can be classified into two types, inhibitory and excitatory. Their collective activity can be derived from the integrate-and-fire models of single neurons by weighting the connection of neurons with coefficients that depend on the strength of the inhibitory and excitatory effect of the synapses. This, in turn, could give some explanation for how decisions are made inside the brain (Chapter 16 in [79]). The activity of collections of interacting neurons is expected to represent information, in the sense of information theory [81], coming from the environment that would be relevant for decision making and functioning of the whole organism. In their seminal paper, Adrian and Zotterman [82] demonstrated that during sensory perception, the intensity of a stimulus is “encoded” by a rate of nerve impulses over time and provided the first conceptual basis for neural coding and neural representations of the sensory worlds [83,84,85]. In addition, it is also thought that inter-spike intervals of individual neurons may also contain some information [86]. However, there still are debates on what is the relevant quantity extracted from the neural activity that is used for functioning purposes of the brain [78]: the “rate code” versus the temporal coding or “spike code” paradigms. For example, it has been shown that sensory information such as taste perceptions are encoded through temporal coding [87]. More recently, spike-timing code was shown to be also critical for motor coordination [88]. While the two views are not mutually exclusives, there are some cases where rate code is not enough for distinguishing external stimuli or where the presence or absence of one spike and its timing is sufficiently informative for inference on the source signal (Chapter 2 in [89]). This debate has been recently considered to be more of a methodological one, and the question of the relevance of the “neural code” metaphor was raised [90,91]. It has been recently found, however, that precise quantification of information available with the neural code could help rule out this debate [92] but it can be challenging experimentally [93], information is by nature dependent on the model of spike activity chosen (Chapter 3 [89]).

### 1.5. Where the “Molecular Brains” Could Take Us?

Of course, the “molecular brains” miss the extreme complexity of networks formed by neurons. However, studying simpler networks may have the advantage of allowing more direct access to the emergence phenomenon. The cellular stage of neural networks and the “black-box” of what occurs in the neurons considerably complicates the understanding of a central question in biology: how signals are propagated and integrated into a network, and how cognitive faculties such as decision making and learning can emerge from it? We infer here that if we look directly at the transfer and processing of information by connected molecules, it becomes theoretically possible to simplify the problem. This perhaps will help to extract what could represent universal principles of emergence in molecular or cellular brains, whatever their scales.

What are the differences and similarities between ribosomal protein networks and cellular signalling networks, and do they have functional and evolutionary relationships? And more generally, are the “informational” molecular networks (permanent or transient) related to nerve circuits? Is there an analogy or even homology between their “information processing” mechanisms at different scales? Or can we imagine that the nervous systems of metazoans and brains have borrowed from cellular networks the components and mechanisms at the origin of their evolution? Could it help to understand how information processing has evolved during major evolutionary transitions (i.e., from unicellular to multicellular life)? And finally, is there an evolutionary link between the mechanisms in molecular networks and in nervous systems? The nervous system of metazoans based on neural networks is well known to perform these sensorimotor functions. While the complexity of metazoan behaviour seems to be related to synaptic plasticity [94], where does the possibility of developing complex behaviour in molecular networks lie? How perception and motricity are integrated into beings without a nervous system remains a fundamental question that has not yet been completely resolved. These questions were asked more than a century ago by Alfred Binet in his book “the psychic life of microorganisms”.

## 2. The Psychic Life of Microorganisms

### 2.1. Cajal and the Neuronal Turn of the Neurosciences

Descartes, who introduced the “machine metaphor of life”, also contributed to the history of the nervous system. In his “Treatise on Man”, he proposed that “information” travels between the brain and the muscles via the nerves. Microscopy confirmed his first observations, and Leeuwenhoek (1695) showed that the nerves coming out of the brain were assembled in hollow fibrils [95]. However, it was only two centuries later that Cajal provided the basis for modern neuroscience, and Sherington would bring us into modern neurobiology, naming “synapse” what Cajal had previously identified as “nerve joint” [96].

Not only by his technical but also conceptual contributions, Santiago Ramon y Cajal was at the base of the neuron theory and modern neurosciences at the end of the 19th century [57,97,98]. By marking them with cell staining techniques developed by Golgi, he showed that all nervous systems are formed by networks of independent cells: the neurons. The meticulous observation of these networks allowed him to elaborate on his famous laws of optimisation of nervous systems: space, matter and time optimisation. In these networks, he also noted in which direction nerve impulses propagate. However, he has already perceived the complexity of the phenomena by noticing that the nervous system also influences the behaviour of peripheral receptors. Thus, he was able to develop an integrative approach that could be considered as the origin of systems biology, which built the foundations of neuroscience [99,100]. A key contribution to his understanding of the nervous system is to have integrated it into an evolutionary context [97,101]. This allowed understanding the general principles of their organisations, from the simplest to the most complex ones. Importantly, Cajal hypothesised the existence of a relationship between the architecture of neural networks and their functionalities, a working basis that will remain until today at the heart of the reflection in attempts to elucidate the functioning of nervous systems [96]. In addition, Cajal’s artistic sense, the quality of his representations and his wonder brought conceptual and technical rigour to science but also blurred the boundaries between science and the arts [102].

### 2.2. Binet, Jennings and Gelber: Toward Neuron-Free Neurosciences?

At the same time in France, Alfred Binet marvelled at another phenomenon that was in some ways close to that of Cajal but at a much smaller scale. “The psychic life of microorganisms” is the title of one of his intriguing works published in 1889 [103], where he describes the great behavioural richness of certain protists under the microscope. Binet, who was also at the origin of the famous IQ test (intelligence quotient) test, could not have foreseen that more than a century later, Michael Galperin would pass it to bacteria and classify them according to their behavioural types [104]. Binet defined two components in “the psychic life of microorganisms”: on the one hand, sensitivity, i.e., the action of the outside world felt by the organism, and on the other hand, motricity, the action of the microorganism on the outside world. Thus, a cell is “irritable because it has the property of responding with movements to the excitations it undergoes”: these are the bases of “cellular sensorimotricity”. He continues: if “psychic life is exercised by a nervous and by a muscular system in the great majority of pluricellular animals”, “it is not the same for microorganisms; most of them have neither a central nervous system nor sense organs; a few do not have locomotion organs”. In his work, he then lists all the organs of locomotion, sense organs that can be observed in protists and notes with relevance the analogies that they can represent with the corresponding organs in metazoans. He observes voluntary and involuntary movements, periodic movements and complex behaviours. Then he describes the complex and quite distinct behaviours that he has been able to observe in different species. He also describes in passing the phototactic properties and sensitivity of bacteria to oxygen tension and about the microorganism nervous system he notes: “we have not found so far in a single proto-organism the slightest rudiment of a central nervous system”. Further on: “it has been said that if there is no anatomically differentiated nervous system in the lower organisms, it must be admitted that their cytoplasm contains a diffuse nervous system”. However, his work, which is of capital importance highlighting phenomena that are still largely unexplained, has long been forgotten. It is likely that the scientific context of the last century and probably “the machine metaphor of life” were not conducive to the birth and development of this theme, which shifts the question of what intelligence is at the molecular scale.

On the other side of the Atlantic, Jennings and Loeb followed in Binet’s footsteps by focusing on the “behaviour” of “lower organisms” [105,106]. Let us note in passing that this categorisation of life forms into “lower and higher organisms” was not conducive to the study of the behavioural complexity of these “lower life forms”. Just as it was unimaginable that continuity could be established between the animal and human world at the time of the publication of “The expression of the emotions in man and animal” [107], it was difficult to conceive at the time when scientists were still trying to demonstrate the superiority of the “white race” by cranial anthropometry [108] that a microbe could be intelligent. Nevertheless, Jennings, very early on, suggested an analogy between the microorganisms’ motility and the sensorimotricity of animals [105]. Half a century later, a decisive step was taken: Beatrice Gelber demonstrated that, like Pavlov’s dogs, paramecia could be conditioned and were therefore endowed with associative learning faculties [109]. Her work was fiercely contested and denigrated for decades, only to be rehabilitated recently in a fascinating review that reopens the question in all its mystery in the light of 21st-century biology [110]: how can beings without nervous systems perform complex behaviour? However, it was not until the 1980s that the behaviour of bacteria entered the disciplinary field of neurobiology [51,52].

### 2.3. Finally, What Is Behaviour?

However, the behaviour is as difficult to define as the brain and is considered as a kind of nebula [111,112,113] whose complexity often escapes the analytical methods of modern science. According to modern neuroscience, a behaviour is dictated by the central nervous system of animals and relies on the activity of neural networks. However, since Binet and Jennings, a growing body of research is converging on the idea that the ability to develop complex behaviours is not exclusive to beings with nervous systems, such as unicellular organisms or plants.

A fundamental and universal behaviour is, first of all, what we could call the “will to live”. Today, these “first wills” have spread, since the origins, to all scales of life and understanding “what bacteria want?” [114] would help to decipher the interplay between the “desire” to survive and the early stages of cell behaviour. One of the most basic forms of behaviour may be the distinct perception of the internal and external world, with the corollary ability to differentiate clearly between the self, other living beings and the physical world [115,116,117]. These faculties are shared at all scales of life: while the sense organs of metazoans are dedicated to the perception of stimuli of all kinds, unicellular beings have numerous sensors that perform an equivalent function but at the micro and nanoscopic scales. “Emotions” that depend on the perception of the world, its attraits or its dangers have probably been intimately interwoven with behaviours since the origin of life. For example, bacteria have soon developed large macromolecular assemblies called “stressosomes”, which are dedicated to reacting to any form of stress. Amusingly, a vast vocabulary from the emotional lexicon is also used to describe the behaviour of ribosomes. For example, one finds in the literature that they can be stressed, resting, taking a pose or even sometimes hibernating [118,119,120]. Moreover, “signalling” is also universal as the genetic code and has evolved from macromolecules to metazoans. Proteins communicate with each other through allosteric processes, using a complex “language” involving different types of mechanisms [121]. Within large macromolecular assemblies such as ribosomes, individual components constantly exchange signals with each other and ribosomes themselves can, by making specific contacts, exchange information [44]. Viruses can make decisions, such as choosing between lytic activity and lysogeny, by communicating through the emission of a chemical molecule, arbitrium [122,123,124,125,126,127]. In a similar way, quorum sensing allows bacteria and other microorganisms to inform each other about their density and trigger collective behaviour [128,129,130,131]. Very diverse modes of communication between cells have also been identified in unicellular beings and metazoans. The exchange of signals can be carried out by the emission of molecules that stimulate specific receptors in the host. However, inter or intracellular communication can also be established in a physical way by electrical, electromagnetic or acoustic waves or mechanical contacts [132,133,134,135,136,137,138,139,140,141,142,143].

In a broad sense, the wide variety of “cellular behaviours” could include the whole repertoire of cellular actions. For example, the metaphor “division of labour” is commonly used for describing the tasks of differentiated cells at the first stages of multicellularity [144]. These may therefore concern not only actions concerning the cell’s reactions to environmental fluctuations but also concerted global actions such as cell division, certain metabolic pathways or even overall cell metabolism and the maintenance of homeostasis. Cell division, for example, is a set of integrated actions ranging from replication to cytokinesis [145,146]. It involves the concertation and synchronisation of a large number of actors. Thus, through the magnifying glass of an ethologist, cell division has all the attributes of what we might call “behaviour”, in the same way as the synchronisation and integration of the stages involved in bird nesting. Taking our reasoning further, if any action that is capable of modulating itself by gauging the outcome of its action in a defined context is “a behaviour”, then the action of an allosteric enzyme can be included in the notion of behaviour. On a slightly larger scale, the steps in the cell’s information cycle that tend towards an integrated “goal” or “function”, commonly referred to as replication, transcription and translation, are themselves units of behaviour integrated into the overall behaviour of ensuring a flow of genetic information in the cell and its progeny. For example, the steps of mRNA translation by the ribosome can be observed in great detail. They represent an extremely complex choreography of gestures and can thus be compared to “a behaviour” requiring sensory-motor control, most probably provided by allosteric networks formed by ribosomal proteins and RNA [44].

### 2.4. Categories of Molecular and Cellular Behaviours: From Sensorimotor Control to Decision Making

At the molecular and cellular level, we can therefore distinguish the first category of behaviour grouping spatio temporal coordination of motions ranging from “simple” phenomena of sensorimotor coordination in the broad sense (molecular or macroscopic) to tropism (global, intra- or extracellular movement leading to a destination). All these phenomena involving space are perfectly regulated in time and coordinated with the aim of global and integrated action. These processes can involve decision making: where to go? depending on several possible choices. A more complex category that includes learning phenomena emerged very early in cellular evolution, which involves decision making based on history. Here, the memorisation of past events plays a key role in the present choice of the organism. Learning, therefore, relies on the ability to remember and recall past actions in order to make decisions. Remarkably, the memorisation of past events already exists in bacteria, which are capable of remembering what has occurred in previous generations [147]. The simplest form of learning is habituation. It consists of attenuating the response and eventually ceasing to respond to a repeated stimulus that has proven not to be harmful to the organism. Habituation was initially well characterised in a mollusc, the aplysia [148], and progressively observed in organisms lacking a nervous system [149]. More elaborate, associative learning makes it possible to establish a link between frequent and joint appearances of different stimuli and to associate them. This faculty is the basis of what is known as Pavlovian conditioning. Growing evidence has shown that they are not exclusive to beings with a nervous system and that complex behaviours such as associative learning and solving complex tasks are observed in unicellulars [110,150,151,152,153,154,155]. Memory is still poorly understood at the neuronal level and is thought to involve synaptic plasticity and the storage of new molecules. At the molecular level, it has been shown to rely on post-translational modification of proteins. For example, the methylation of bacterial chemoreceptors or the phosphorylation of receptors is thought to be involved in cell memory and signalling. What occurs in molecular networks is therefore crucial for understanding cellular behaviour and the emergence phenomena leading to the notion of life.

### 2.5. Cell Cognition and Consciousness

The cause of the intelligence of plants has moreover joined that of bacteria, and these questions of status in the scale of the living have given rise to heated debates [156,157,158]. In addition, Galperin, who had the audacity to measure the IQ of bacteria and to propose, on the basis of genomic studies, very distinct behavioural types, has given rise to many controversies that the suitable reductive session of modern, factual biology has found difficult to accept [104,114]. After this long period of maturation, we are today witnessing a flowering of articles that tend to demonstrate the universality of the notion of consciousness, self-perception, cognition and intelligence in organisms or colonies that they can form, whether or not they have neurons [11,26,159,160,161,162,163,164,165,166]. These recent studies are all the more interesting as they place the concept of intelligence in an evolutionary context and propose very old roots of nervous systems that could have originated in systems performing similar functions but at the cellular level [59,167,168,169,170,171,172,173,174]. Moreover, a recent study proposes to link the cognitive faculties of microorganisms in an evolutionary and ecological perspective [175]. This work would make it possible to appreciate the evolution of “intelligence” in an environmental context and to have a global vision of the factors that may have enabled its development. What we observe a century after Binet’s “psychic life of microorganisms” goes far beyond what he could have imagined. The understanding of these phenomena has moved to the molecular level, and several decades of “cell signalling” studies have provided the fundamental principles and the molecular mechanisms that govern cellular signal transmission, integration and decision making.

## 3. Cell Signalling and Sensory Motricity

Cells have the ubiquitous ability to perceive, integrate multiple (external or internal) stimuli and make appropriate decisions that allow them to maintain homeostasis and choose survival strategies in fluctuating environments. Several decades of study in both structural and systems biology have led to a considerable evolution in the overall understanding of cell signalling on the one hand and in the detail of the molecular mechanisms of signal transduction on the other [41,42,43,176,177,178]. While the basic schemes of cell signalling are similar in prokaryotes and eukaryotes, they involve distinct components and signalling pathways [179,180]. In prokaryotes, the most common systems are the 1-component systems consisting of a protein that is both the receptor and the effector acting on gene expression [181]. In 2-component system signalling, the task differentiation between the receptor, a sensory kinase and a response regulator corresponds to a later evolutionary stage [182,183,184,185,186,187,188]: the signal transduction is accomplished by the transfer of a phosphoryl group from a histidine of the sensor protein kinase to an aspartate of a response regulator. Variations around the theme of “phosophorelay” and post-translational modification have subsequently evolved to provide signalling cascades in eukaryotes. Although increasing complexity of signal transduction mechanisms has accompanied major evolutionary transitions (the transition from prokaryotes to eukaryotes and the development of multicellularity in eukaryotes), cell signalling has followed complex evolutionary paths mixing convergence and divergence from common molecular components [189,190,191]. These systems combine distinct mechanisms for triggering and propagating signals: (1) allostery, (2) protein oligomerisation and partial unfolding, (3) reversible chemical or post-translational modifications [192], (4) production of a great variety of short-lived “second messengers” (small molecules and ions), (5) abrupt changes in the membrane potential, through the opening of membrane ion channels, (6) flows, of electrons, protons or photons. These different types of signals must be able to inter-convert and “understand” each other in order to establish a chain of communication between different actors in the cell signalling processes. There is, therefore, a process of “translation” between the different informational codes in signalling.

On the other hand, cells have learnt how to turn the jiggling and wiggling of atoms [193] into a harmonious choreography. It is indeed fascinating to see how the information contained in the biological macromolecules has made it possible to constantly play between disorder and order to install a harmony where the movements of the actors seem perfectly synchronised. How, at each scale of time and space, is this synchronisation, which one might be tempted to compare to a sensory-motor synchronisation, achieved? How to create orderly and concerted movements in the stochastic universe of thermal fluctuations has been one of the major challenges of evolution through the interplay of the bio-polymer sequence. One of the first autonomous movements that a protein sequence accomplishes when it leaves the ribosome is to fold. While co-translational folding is most often assisted by the ribosome [194,195], the protein sequence generally contains the information necessary for well-coordinated folding [196,197,198,199]. This allows it to make use of thermal fluctuations while also thwarting the purely stochastic aspect of Brownian motion. Once folded, each protein moves and vibrates according to its type of folding, its sequence and its partners [200,201,202]. There is then a gradation of movements from the dynamics of a single protein to the concerted and synchronised movements of a large number of partners in macromolecular complexes during replication, transcription and translation [203,204,205]. Moreover, all these processes are coupled and interdependent.

While the cytoplasm is inherently subject to the vagaries of molecular diffusion, cells have developed different ways to control and even amplify these processes [206]. However, in all three major kingdoms, cells have developed filaments specifically dedicated to controlling cell shape and cytoplasmic movement. These “cytomotor” filaments use energy from GTP or ATP to control their directional assembly and thus create forces used to control cell shape, move organelles or organise membrane systems [207,208,209]. There are essentially two protein classes of cytomotor filaments: actin filaments, tubulin filaments and intermediate filaments that make connections between them. Each of these filaments is also associated with numerous accessory proteins that give them particular motor properties. In eukaryotes, for example, actin filaments are associated with myosin to exert forces between them, allowing movements responsible for cytokinesis, modification of cell shape or transport of organelles [210,211,212,213,214]. Tubulins associated with dyneins are responsible for chromosome movement during mitosis [215,216,217,218].

Cell motility evolved independently in all three kingdoms, either from the motile properties of cytoskeletal elements or from distinct proteins [219,220]. Archaea have evolved archaella, bacteria flagella [221] and eukaryotes cilia [222,223,224]. We are beginning to understand in detail the molecular mechanisms that control cilia beating, which involve around 100 players [225,226]. Cellular signalling establishes the equivalent of sensory-motor coordination between these “locomotor organs” and the various sensors (see above) to define an appropriate behavioural response.

### 3.1. The “Sensitive” Nature of Biological Molecules

Both prokaryotic and eukaryotic cells possess a huge variety of molecular sensors. From simple receptors to large complexes such as stressosomes or magnetosomes to RNA riboswitches, these sensors provide cells with precise and detailed information on fluctuations in their environment or intracellular compartment. However, due to their sequences that provide an infinite diversity of tertiary interactions that tune and maintain their three-dimensional folding, biological macromolecules are inherently sensitive. A simple rise in temperature or the binding of a ligand can lead to structural or dynamic modifications that will change its structure and properties or allow a disturbance to be propagated. Evolution has exploited the sequence-encoded subtle interplay of intra- and inter-molecular interactions to generate sensors that are responsive to almost any possible physical or chemical stimulus that a cell may perceive. Many molecular mechanisms used by proteins and nucleic acids to sense different types of physical or chemical stimuli are now deciphered.

### 3.2. Gated Channels

The first category of sensors is represented by ion channels, which can convert and amplify any type of signal into abrupt changes in cell electrical activity. Several families of ion channels whose opening is controlled by different stimuli, including ligands, protons or physical stimuli such as voltage, have been listed [227].

#### 3.2.1. Voltage-Gated Channels

For example, voltage-gated ion channels may open or close in response to changes in the membrane potential and play a key role in electric signalling from bacteria to vertebrates [228,229,230]. K^+^, Na^+^ and Ca^++^ voltage-gated channels share the same four-fold symmetric architecture consisting of a voltage-sensing domain located in the periphery and a central pore-gated domain that confers the selectivity for different cations [231,232]. The voltage-sensing is performed by a well-defined and highly conserved structural domain consisting of four transmembrane helices whose assembly changes conformation according to the membrane electric field. The S4 helix contains “gating charges”, positively charged amino acids that delocalise in response to the membrane electric field [233]. Structures of voltage-gated channels at different functional states suggested that the movement of gating-charge is coupled to the opening of the pore-gated domain, which adjusts its conductance according to the voltage [234,235,236]. This canonical model, however, may display some variations as shown by recent structures and molecular simulations [237,238,239]. Interestingly, the voltage sensor domains have evolved to couple voltage-dependent conformational changes in a variety of functions and act as modular units that confer voltage-sensing to other enzymes, such as phosphoinositide phosphatase [240,241,242,243]. On the other hand, other structural motifs involving different mechanisms, such as dipole motions, have also been found to confer voltage sensitivity in proteins [244,245].

#### 3.2.2. Temperature-Gated Channel

The opening of some channels, such as some members of the TRP family, can be controlled by temperature [246,247]. The control mechanisms are more complex than voltage since it is frequently coupled with other stimuli, however. Temperature sensitivity may involve either a global response of the protein or a domain specifically dedicated to temperature perception. While T° sensitivity is an inherent property of proteins and therefore can be globally sensed by the whole protein, some structural motifs have evolved to provide a specific response to temperature changes. Studies have shown that the sensitivity of certain regions to temperature may depend on their degree of disorder. For example, it has been suggested that the more dynamic regions of TRPV1 channels have a higher thermal sensitivity and facilitate the uptake of energy from its surroundings and may reciprocally transfer it to neighbouring secondary structures, such as β-sheets that have different thermal properties to induce an allosteric response [248,249]. Thus, according to Diaz-Franulic, TRPV1 channels that exhibit polymodal responses to different stimuli [250,251] could transmit an anisotropic thermal response from one domain to another [252,253]. On the other hand, specific modules can also confer temperature sensitivity to certain channels, such as the unfolding of a temperature-sensitive domain of the BacNav channel [254,255]. More complex mechanisms are observed in some channels where temperature acts by altering the coupling between the calcium-sensitive domain and the Ca^++^-dependent pore domain of the archaeal MthK channel [256].

#### 3.2.3. Mechanosensitive Ion Channels

Ion channels sensitive to mechanical stimuli play an important role in mechanotransduction by controlling the opening of channels in membranes along which mechanical forces are exerted both in bacteria [257,258,259,260,261,262,263] and in specialised cells in eukaryotes [264,265,266]. These channels are often associated with the perception of other stimuli such as voltage or ligands. Although the structures of mechanosensitive channels in pro- and eukaryotes are different, similar mechanisms have converged to perceive membrane deformations and couple them to pore opening to trigger permeation. Lipid bilayer tension helps trigger channel opening. Bacterial MscS (small conductance) and MscL (large conductance) mechanosensitive channel structures consist of a homo-heptamer and a large cytoplasmic domain [267,268,269]. Differences in membrane bilayer thickness associated with voltage changes induce a change in the tilt of the TM1-TM2 helices, which is then coupled to pore opening [269]. In addition, this study also shows how the acyl chains of the lipid gatekeepers contribute to stabilising the pore in its closed state. These dissociate under membrane tension. In eukaryotes, mechanosensitive piezo channels form huge structures of more than 2500 amino acids containing 38 transmembrane helices with an overall three-bladed, propeller-shaped trimeric architecture. These structures suggest that these long blades can couple membrane deformations to pore opening via a lever arm mechanism and allow selective permeation of cations [270,271,272]. On the other hand, mechanosensitive K^+^ TRAAK channels, members of the two-pore domain K+ (K2P) family, also show that lipids and mechanical deformations of membrane segments induced by membrane tensions participate in the control of pore opening [273].

#### 3.2.4. Light-Sensitive Channels

Light-sensitive ion channels also exist in some unicellular algae that participate in phototaxis in green algae (chlorophytes) by depolarising the plasma membrane [274,275]. These channels, channelrhodopsins (737 aas), evolved from bacterio-rhodopsin consisting of seven transmembrane segments covalently linked to a retinal chromophore [276]. Light absorption induces isomerisation of the retinal, which, in turn, causes a set of conformational changes leading to the opening of a pore that allows ions to pass through [277,278,279]. In cryptophytes, a distinct group of green algae, there are also cation-conducting channel rhodopsins that specifically conduct anions [280,281] and another group of channels that are more structurally related to haloarchaeal rhodopsins and have different functional properties [282]. Light-sensitive channels have also been found in nucleocytoplasmic large viruses that infect plankton [283].

#### 3.2.5. Ligand-Gated Channels

There are also channels whose opening is controlled by ligand binding, of which there are several families such as pentameric ligand-gated channel, glutamate-gated channel, K_ATP_ channels [284]. In addition, some channels whose opening is triggered by physical stimuli can also be controlled by specific membrane lipids [285]. Although these families have different architecture and evolved from different structures, they share a modular organisation of intra- and extra-membrane domains that allows the conversion of ligand binding into an electrical signal by controlling the opening of transmembrane ion channels. These mechanisms can be finely modulated by a series of allosteric modulators. Thus, while binding of “agonist” ligands to orthosteric sites triggers pore opening, binding of allosteric modulators to distinct sites can modulate the response to orthosteric ligands [286].

*Pentameric ion channels (pGLIC)**.*** The pGLIC or pentameric ligand-gated channels represent a ubiquitous family found in bacteria, archaea, and eukaryotes [287,288,289,290]. They consist of a symmetric or pseudo-symmetric assembly of five subunits surrounding a central pore selective for cations or anions [291,292]. A recent phylogenetic study has shown that these channels fall into two clades, those that share a loop (cys-loop) whose structure is maintained by a disulfide bridge between highly conserved cysteines and their metazoan, unicellular eukaryotic, and prokaryotic orthologs where the cys-loop is not conserved [288]. The structures of the homo-pentameric prokaryotic proteins have been solved at high resolution and provided detailed insights about their activation mechanisms [293,294]. Although sharing the overall architecture and activation mechanism of their eukaryotic counterparts, they display simpler behaviours than their eukaryotic homologs. For example, the *Gloeobacter violaceous* pGLIC homolog is a proton-gated channel thought to contribute to adaptation to pH changes [295]. In metazoan, pGLIC play a central role in the central and peripheral nervous system, and their mechanisms have been studied in detail due to their pharmacological applications. In synapses, they convert a chemical signal induced by a neurotransmitter released from the pre-synaptic membrane into an ionic flux in the post-synaptic membrane. There are channels that respond to excitatory neurotransmitters (Ach, 5-HT) that control cation flow (Na^+^, K^+^, and Ca^++^) [296], and inhibitory ones that control anion channels (GABA, glycine, and glutamate in invertebrates) [297,298,299,300]. Decades of biochemical, mutational, and structural studies have provided insight into the molecular mechanisms of the coupling between ligand binding to the extracellular domain, and transmembrane channel opening, and the modulation of the process by allosteric modulators located at distinct sites [296,301,302,303,304,305,306,307,308]. Interestingly, the orthosteric ligand binding (e.g., Ach) involves numerous aromatic residues that form cation-π interactions with it [309,310].

*Glutamate-gated channels*. Glutamate-gated channels are the main players in the excitatory synaptic response [311,312,313,314,315]. Although the three AMPA, kainate and NMDA subfamilies differ in their response times, the nature of their allosteric modulators, and their ability to desensitise, they share a common architecture, consisting of homo (kainate and AMPA) or heterotetramers assembled into a dimer of dimers. Each subunit is composed of an amino-terminal domain (ATD) involved in receptor assembly and localisation, a ligand-binding domain (LBD) and a transmembrane domain whose structure is similar to K^+^ voltage-gated channel. The structure of the whole AMPA receptor confirmed that, contrary to cys-loop receptors, ligand binding occurs on each subunit (clamshell) and not between the subunits. Ligand binding there induces closure of the LBD clamshell [316] and movement involving the M3-S2-M4 region that causes a conformational change leading to pore opening [317,318]. Interestingly, as in the case of cys-loop receptors, inter-subunit cation-π interactions could participate in the allosteric mechanism.

-*K_ATP_ channels.* ATP-sensitive K channels (K_ATP_) are the molecular sensors of cellular metabolism. They control the opening of the channel according to the intracellular concentration of ATP [319,320]. They convert the metabolic state of the cells (ATP/ADP ratios) into an electrical signal. These channels form hetero-octamers consisting of four Kir6.1 or Kr6.2 domains structurally close to the inward rectifier Kir channels family and four sulphonylurea receptor domains (SUR1 or SUR2A/B), members of the ABC transporter family. Three natural ligands control its activity: PIP2, ATP and ADP. While PIP2 is required for its activity (opening of the channel, ATP exerts an inhibitory action by binding to the Kir subunit, which induces channel closure). On the contrary, the binding of ADP on the SUR domains potentiates its activity by promoting opening. Recent structures of whole K_ATP_ channels have identified ligand-binding sites and elucidated molecular mechanisms that establish structural interconnection between Kir and SUR domains that may explain the intricate mechanisms of ligand actions on pore opening control [321,322].

### 3.3. Membrane Receptors

In addition to gated channels, a great variety of membrane receptors sense all kinds of physical and chemical stimuli. They relay and convert them in various cellular responses such as protein phosphorylation, synthesis of second messenger and the interconversion of GDP into GTP. They participate in the complex and intricate networks of cell signalling that has been largely documented during the past decades.

#### 3.3.1. G Protein-Coupled Receptors (GPCR)

G protein coupled receptors are the largest and most ubiquitous cell surface receptors in eukaryotes that transmit signals induced by a broad spectrum of ligands, from photon to large protein molecules, via heterotrimeric GTP binding proteins (G-proteins) or β-arrestins [176,323]. Members of this family are involved in diverse signalling pathways and include light-receptors such as rhodopsin, receptors for ions, neurotransmitters, hormones, growth factors, chemokines, as well as sensory receptors for various odorants [323]. Approximately 2% of the human genome encode members of the GPCRs family that regulate key physiological functions such as sensory perception, neurotransmission, immune responses, developmental processes, regulation of endocrine and exocrine gland functions, glucose and lipid metabolisms, blood pressure. While they were initially considered as cellular membrane receptors, numerous studies have shown that GPCRs also signal at various intracellular locations using a wide variety of signalling modes and mechanisms [19]. All GPCRs are integral membrane proteins that share a structurally conserved domain composed of seven transmembrane α-helices that cross the membrane, thus forming three intracellular and three extracellular loops [323,324]. They also contain variable extracellular amino terminus and intracellular carboxy terminus tails. GPCRs are now thought to have a common ancestor with sodium-translocating microbial rhodopsin [325,326,327]. They were already present in the unicellular eukaryote ancestors and have been grouped in several classes (A-F) according to their sequence similarities and regulation and ligand-binding modes [328,329,330,331] (https://GPCRdb.org. 18 October 2021). Class A (*rhodospsin*) is the larger group and includes the well-characterised rhodopsin and adrenergic and olfactory receptors. Three subtypes differ by their ligand-binding modes. In subtype 1, the ligand-binding site is deep within the transmembrane domain. In subtype 2, ligands bind to aminoterminus and extracellular loops. In subtype 3, the ligand-binding involves a long extracellular domain. The class B receptors (*secretin*) are activated by high-molecular-weight hormones, and the class C (*glutamate*) ligands bind on a very long terminal domain, and activation involves obligate dimerisation. Other classes include the class D (fungal mating pheromone receptors), class E (cAMP receptors) and class F (Frizzled receptors). An intriguing finding is that GPCR-mediated neurotransmission homologs already exist in primitive nervous system and nerve-cell-free organisms [332].

Ligand-binding stabilises an active receptor conformation that couples GPCRs to heterotrimeric G-proteins (Gαβγ) and promotes the exchange of GDP for GTP in the Gα subunit. The nucleotide-binding pocket of the Gα subunit is located between the Ras-like domain and the α-helical domain [333]. GTP binding then dissociates the Gα-GTP subunit from the dimeric Gβγ subunit. Both Gα-GTP and Gβγ subunits can then independently regulate a variety of downstream effectors that, in turn, stimulate various cellular responses depending on the G protein-coupling specificity of each receptor. The GTPase activity of the Gα subunit induces the hydrolysis of GTP to GDP and the subsequent reassociation of Gα-GDP and Gβγ subunits, which makes them available again for the next cycle. Based on sequence homologies of their Gα subunits, the G-proteins are grouped in four distinct families Gα_s_, Gα_i_, Gα_q_ and Gα_12_ that activate distinct cell signalling cascades. For example, while Gα_s_ promotes the activation of adenylyl cyclases that convert ATP to the second messenger cAMP, Gα_i_ inhibits adenylyl cyclases and decrease the cytosolic level of cAMP. Gα_q_ activates phospholipase Cβ that produces the second messenger’s diacylglycerol (DAG) and inositol triophosphate (IP3) [17,334]. The binding of a single ligand to GPCRs can lead to the activation of several G-proteins and thus constitute the first step in signal amplification. Then, activated G-proteins trigger the production of second messengers that target a large number of ion channels, calcium-sensitive enzymes and various kinases that contribute to relay the signalling cascade into the cell. GPCR kinases (GRK) terminate the signalling cycle by the GPCR phosophorylation that promotes their binding to β-arrestins and receptor internalisation, recycling or degradation [19].

Since the first high-resolution structures of the β2 adrenergic receptors-Gs complex [335,336], the structural studies on GPCRs and GPCR-transducer complexes have provided considerable insights into their ligand-activation and allosteric mechanisms of G protein-coupled receptor activation (reviewed in: [17,334,337,338,339,340,341]). Allostery reciprocally modulates the behaviours of GPCRs and the G-proteins: ligand binding on the extracellular side of the receptor promotes the G protein binding on the GPCR intracellular side, and the G protein increases the GPCR affinity for the ligand. Common structural features have been found in the allosteric process that couples agonist binding in the GPCR-A class to G protein binding and activation. The common pathway involves TM6 movement and key conserved motifs and notably conserved aromatic acid residues that link the ligand-binding pocked to the G protein coupling region [342,343]. These studies have also contributed to the evolution of the concept of GPCR signalling from simple “on/off switches” to a more complex system and led to a change in the paradigm of GPCR activation. Indeed, most receptors exhibit a basal level of GTP exchange activity even in the absence of ligand [16,344]. On the other hand, the loose coupling observed between ligand binding and G protein or arrestin interaction has [17,345,346] indicated that the agonist does not simply stabilise the receptor in an active conformation. The receptor is in a preexisting equilibrium between inactive and active conformation, and the ligands shift the population ensemble of preexisting conformations rather than stabilising the unique activated state.

#### 3.3.2. Kinases and Protein Phosphorylation

Reversible protein phosphorylation is universally involved in cell signalling processes and often follows a recurrent logic that involves the concerted action of three concerted actors: “writers”, kinases that phosphorylate specific sites in proteins, “readers”, modules that recognise and bind to phosphorylated sites, and “erasers”, protein phosphatases that remove phosphate groups from phosphorylated proteins [189]. The modularity of the organisation of these three players provides a wide range of regulatory mechanisms and has played an important role in cell-cell interactions and in the emergence of metazoans. While in bacteria, the main players are histidine kinases, in eukaryotes, the writers are mostly serine/threonine kinases (STKs) and tyrosine kinases (TKs).

Protein phosphorylation leads to two (non-exclusive) events that can produce a cellular response: it can produce a conformational change that propagates through an allosteric pathway. Another way is to induce the binding of protein modules that specifically recognise phosphorylated sites [347] and trigger a cascade of events leading to specific signalling in the cell.

In eukaryotes, two signalling systems have evolved [348]: (i) Ser/thr kinases: ser and thr represent 97% of the phosphorylated amino acids in proteins and induce conformational changes when modified. (ii) Tyrosine kinases act by inducing protein-protein interactions. By specifically recognising phosphorylated tyrosines, SH2 domains spatially and temporally control proteins that contain phosphorylated tyrosines. SH2 domains have a well-conserved folding structure consisting of a central 3-stranded antiparallel β-sheet flanked by two helices. These domains, which form modular associations with other domains in effectors or in the protein kinases themselves, specifically recognise phosphorylated tyrosines (pYs) in multiple sequence contexts, a process governed by a subtle code that brings together the structural, dynamic, thermodynamic and kinetic properties of the partners’ association. The mode of pY binding is universally conserved, whereas recognition of the surrounding sequence is more variable and is responsible for the specificity of SH2. These modules allow circumstantial regulation and integration of multiple signals [347]. The receptor phosphotyrosine kinases (RTKs) display an extremely diverse modular organisation. They consist of a variable extra-cytoplasmic receptor, a transmembrane helix and an intracellular catalytic domain [349]. The signalling process takes place in several sequential steps, usually consisting of the stimulation of the extracellular domain by a ligand that induces receptor dimerisation and kinase autophosphorylation that contributes to increasing its catalytic activity and the binding of effector proteins containing SH2 or PTB domains that activate a cascade of signalling events induced by phosphorylation of other sites on downstream targets and nucleation of signalling complexes [350,351]. This process, which links pY to the modular binding of many effectors, also allows orthogonal signalling of multiple signalling systems [189]. Receptor activation can also stimulate the formation of receptor clusters (somewhat analogous to chemotaxis receptor clusters in bacteria) whose cooperative association can produce complex signalling.

#### 3.3.3. Photoreceptors

The light-induced behavioural response of microorganisms is mediated by photoreceptors [352,353,354,355,356,357,358,359,360,361,362] and mediates either non-directional photoreception that monitors variations in the light intensity or directional photoreception that is used for orientation and habitat selection. High-resolution vision constitutes the most elaborate system that enables complex behaviours such as prey detection [353,354,363]. Various systems have convergently evolved to provide accurate light-sensing such as the cellular microlenses of *Synechocystis* cyanobacteria used as micro-optic to sense light direction [364] or the elaborate eye-like ocelloids of certain dinoflagellates [365]. The best-characterised families of photoreceptor proteins belong to different families and folds and are associated with distinct chromophores [366]. Despite their diversity, they share a similar property, which is to convert the ultrafast local structural photoactivated changes in their pigment into long-lived global changes in the receptor protein that are transmitted to other proteins in the signalling chain [352,360]. The rhodopsin and photoactive yellow protein (PYP) use the photoisomerisation of a C=C bond in the retinal or the *p*-coumaric acid, respectively. Phytochromes use the bilin, an open tetrapyrrole and the LOV proteins, cryptochromes, and BLUF proteins use the flavin mononucleotide (FMN) as chromophore [367].

The rhodopsins consisting of seven transmembrane α-helices belong to the vast family of G protein-coupled receptors [324,368]. The light-induced isomerisation of the 11-cis-retinal is coupled to a cascade of events that relay the signal to guanylate kinases [369,370] or phosphodiesterase [371] that trigger various intracellular responses. The PYP adopt the fold of the Per-ARNT-Sim (PAS) domain [372,373] that senses a vast range of stimuli, from photon to ligand [374]. The *trans* to *cis* photoisomerisation of its *p-coumaric* acid chromophore induces small structural rearrangements of the protein [375,376] and is associated with charge delocalisation around the chromophore [377]. The ubiquitous LOV (light, oxygen or voltage) photosensors domains also adopt the PAS domain structure, but they use the FMN as a chromophore [356,378], and they are associated with various effectors such as histidine kinases, protein involved in synthesis of cyclic cGMP, STAS (anti sigma antagonist), helix-turn-helix. They have also been found to be associated with other sensor domains [379]. Several mechanisms of signalling have been proposed on the basis of their dark and light-activated structures, such as helix Jα unwinding [380,381] or dimerisation [382]. BLUF, bacterial and eukaryotic blue light receptor using flavine adenine dinucleotide (FAD), has a modular architecture comprising a 150aas receptor domain with a ferredoxin-like fold that can be connected with many different effectors [383,384,385,386,387]. They can be found as the single photosensory domain involved in light-regulated protein interactions [388] or in multidomain proteins fused to light-regulated transcriptional effector [389,390], phosphodiesterases [383] or adenylyl cyclases [391,392]. BLUF domains bind FAD/FMN/RF pigment non-covalently for the properties of the isoalloxazine to absorb blue light. In BLUF, a photo-induced proton-coupled electron transfer (PCET) initiate a rearrangement of hydrogen bonds around the flavin cofactor after illumination, which is transmitted to the surface of the receptor and lead to the activation of the various effectors. Cryptochromes adopt a Rossman-like fold similar to photolyases [393,394] involved in functions ranging from DNA repair to blue light regulation of growth, development and circadian rhythms [395]. Cryptochrome photoreception is based on blue light-induced interconversion between several redox states of flavin as chromophore [396]. While the role of the tryptophan triad is controversial [397], the light-induced oligomerisation of plant cryptochrome is an accepted mechanism for light-regulation interactions with various signalling partners [398,399,400]. Phytochromes are modular multidomain red/far-red photosensory proteins that share conserved PAS, GAF and PHY core photosensory domain that adopt a knotted structure [401] and bind covalently a linear tetrapyrrole chromophore (bilin, phytochromobilin, or biliverdin) in a broad range organisms such as bacteria, unicellular eukaryotes and plants [402,403,404,405]. Photoconversion triggers structural changes in the dimer interactions and the refolding of a “tongue” loop that modulate the activity of C-terminal “output” domains [406,407].

### 3.4. Other Molecular Sensors

#### 3.4.1. Magnetoreception

Magnetotactic bacteria have, for example, evolved magnetosomes to perceive and orient along the Earth’s magnetic field [408] and lead to complex collective phenomena of some bacteria subjected to a magnetic field [409,410]. These organelles consist of magnetite (Fe_3_O_4_) wrapped in cell membranes organised along the motor axis by cytoskeleton proteins [411,412]. These structures are encoded by conserved gene clusters that have diverged in a large number of magnetotactic bacteria [413]. However, the process may be more complex than expected as a link has been established between the magnetotactic properties of some bacteria and aerotaxis-dependent signal transduction systems [414]. Proteins involved in the magnetotactic response whose genes are in close proximity to genes involved in signal transduction are beginning to be characterised structurally [415]. In animals and among them butterflies, other systems appear to be involved in magnetic field perception, such as radical pairs and cryptochromes [8,9,416,417].

#### 3.4.2. Stress and Stressosome

Another important signalling complex has been discovered in some bacteria: the stressosome [418]. This complex is formed by the pseudo-icosahedral assembly of RsbR (sensor), RsbS (scaffold) and RsbT (kinase) proteins [419,420]. This complex responds to environmental changes that could be sources of stress such as ethanol, UV or osmolarity by triggering cascade events leading to the expression (release) of alternative sigma factors (sigma B), which, in turn, activates over 150 stress-related genes. The molecular details of this phosphorylation cascade are not yet well understood.

#### 3.4.3. Sensing with Nucleic Acids

Beyond protein receptors, single-cell and pluricellular organisms have developed a great variety of sophisticated systems to monitor and respond to temperature by different molecular strategies [421,422,423]. For example, with their nucleic acid thermosensors, also called “RNA thermometers”, bacteria register temperature changes by using temperature-modulated structures in the untranslated region of some mRNAs. Temperature sensing is based on several non-canonical, heat-labile base pairs temperatures [424,425,426,427,428]. Conversely, cold sensor group II introns are self-splicing ribozymes where cold-induced disruption of key tertiary interactions prevents splicing and triggers various cellular responses [429]. In addition, RNA riboswitches may also sense a variety of metabolites [430,431,432,433,434,435,436] and a great variety of protein chemosensors involved in chemotaxis.

However, molecular sensing can also be extended to other molecular mechanisms and functions as diverse as the control of correct pairing during replication by DNA polymerases or the detection of node chirality by type II topoisomerases [437,438,439]. Cells can also sense and control the DNA integrity in monitoring the long-distance migration of charges through the aromatic base-pair stack within the DNA helix [440,441]. In addition, there is growing evidence that many reactive oxidative and electrophilic species (ROS/RES) act as cellular signals and can mediate inter-organelle or intercellular redox information exchange [442,443].

### 3.5. The First Proto-Brains’ Ideas in Bacterial Chemotaxis

Despite their functional analogies, cellular signalling and nervous systems have long remained two very distinct disciplines. Even today, few bridges are established between these two themes. The first daring analogies between cell signalling and neurobiology emerged with bacterial chemotaxis [51,52]. Still, these articles have remained surprisingly little cited. Bacterial chemotaxis was thus the first step, in the 20th century, of a fundamental question: how can a bacterium orient itself, memorise and become part of history, learn and even decide where it will go according to chemical stimuli that repel or attract it in its environment? We began to realise that bacteria could make decisions in complex situations [444,445], amplify signals, show habituation faculties and retain the memory of past situations [446,447]. Chemotaxis allows the bacterium to choose its orientation according to a source of attractive or repulsive substances [445]. It is integrated into a broader context of perception of external or internal signals based on a principle of signal transduction: two-component systems [448]. In these sophisticated 2-component systems, a “sensor” protein auto-phosphorylates in response to a specific signal coming from specific receptors and transfers the phosphoryl group to a “response regulator” protein that carries out a cellular response [114,449,450,451,452,453]. The constant comparison of the signals perceived at a past and a present moment decides the direction of rotation of the flagellum [454,455]. When the flagellum rotates clockwise, the bacterium oscillates in a kind of random walk. When the flagellum rotates counter-clockwise, the bacterium moves in one direction only [456]. This alternation between “run and tumble” thus allows it to orient itself along a chemical gradient. The last decades have provided detailed information on the molecular mechanisms responsible for this sophisticated behaviour. They have identified most of the bacterial chemotactic receptors, sensor and response regulator proteins associated with signal transduction. The structure of individual components and clusters of receptors grouped in networks has also been resolved [457,458,459]. These clusters formed by the assembly of various types of receptors have the property of amplifying and integrating all signals. Thus, their cooperation in response to a given chemical signal makes it possible to generate an appropriate response, even in contradictory situations [444]. Their sensitivity is modulated according to what they have previously perceived through the methylation of a protein region (HAMP), which thus keeps the memory of what the bacterium has encountered. These networks of receptors and sensor and regulator responses were, therefore, the first molecular assemblies to have been compared to proto-brains [48,49]. They confer to each bacterium an individual behaviour that varies according to its own history and the different situations it may have gone through [147]. Interesting data have also been produced by genomics and structural studies, which have shown that the organisation of these clusters is universal in bacteria [460]. Although the diversity of receptors for various substances may vary from one species to another, the basic organisation and mechanisms involving two-component systems are common in all chemotactic bacteria [104,461,462]. Thus the networks of the component of bacterial chemotaxis with emergent properties, allowing bacteria to develop complex behaviours, were the first to be compared to molecular brains, which are moreover frequently found at the “head” of bacteria. As seen above, in his 1995 seminal paper, Dennis Bray proposed one of the most exciting hypotheses in biology: circuits formed by protein networks may play a role analogous to the nervous systems in cells [13]. Although the emergence of these “proto-brains” probably coincided with the beginning of life, it took another two decades before they were explicitly named [48,49].

## 4. Ribosome Signalling and Primordial Molecular Brains

Another analogy with the nervous circuit has been proposed for the r-protein networks observed in the ribosome [44,45]. Interestingly, although both ribosomes and chemoreceptor arrays have been compared to brains, they differ significantly in their symmetry and geometric properties: while chemotactic receptor arrays form regular and symmetrical structures, r-proteins display long unstructured regions and form totally irregular networks (Figure 3). Yet, although asymmetric, the ribosome is considered as a window towards the earliest forms of life that predate the three kingdoms and therefore, the analogy of the ribosome with brains can provide interesting information on both the functional and evolutionary levels.

While in astrophysics looking far away gives the opportunity to glimpse the fossil radiation of the universe, looking into the heart of the ribosome may tell us what the first forms of life might have looked like. The ribosome evolved by accretion around a core that predates the radiation of the three kingdoms and were probably present in LUCA (Figure 2) [463,464,465,466,467]. The ribosomes are thus considered as a relic of ancient translation systems that co-evolved with the genetic code have evolved by the accretion of rRNA and ribosomal (r)-proteins around a universal core [466,468,469,470,471,472]. They then followed distinct evolutionary pathways to form the bacterial, archaeal and eukaryotic ribosomes whose overall structures are well conserved within kingdoms [55,473,474,475]. The complexity of ribosome assemblies, structures, efficiencies and translation fidelity concomitantly increased in the course of the evolution.

Strikingly, with their long filamentous extensions, ribosomal proteins (r-proteins) form neuron-like networks that “innervate” the ribosomal functional centres such as tRNA sites, PTC, and the peptide tunnel [44,45]. In these networks, r-proteins are interconnected by the formation of molecular “synapses”, very small and phylogenetically conserved interfaces between the filamentous extensions of the ribosomal proteins that thread their way through the interstices of the rRNA [45]. These tiny structures are stabilised by their interactions with the rRNA and display a “necessary minimum”, conserved aromatic-basic amino acid motifs that are also shared by larger interfaces. It has been proposed that these highly conserved interfaces have been selected during evolution to play a specific role in inter-protein communication, and they reveal the strictly necessary interacting residues to ensure information transfer from a protein to another (Figure 4). An interesting hypothesis is that these minimalist “molecular synapses” reveal much more general principles in molecular communication. Indeed, these tiny interfaces, which appear in their simplest expression in the ribosome thanks to the spatial constraints of ribosomal RNA (rRNA), could be ubiquitous in macromolecular complexes but drowned out by a “structural” background involving other amino acids for their stabilisation.

The r-protein networks could contribute to both the ribosomal assembly and the “sensorimotor control” of protein synthesis. Many experimental studies have indeed shown indeed that ribosome functional sites continually exchange and integrate information during the various steps of translation. As the numerous studies of the Dinman group have shown: “an extensive network of information flow through the ribosome” during protein biosynthesis [476,477,478,479,480,481,482,483]. For example, several studies have also demonstrated long-range signalling between the decoding centre that monitors the correct geometry of the codon-anticodon and other distant sites such as the sarcin ricin loop (SRL) or the E-tRNA site [473,484]. R-proteins of the ribosomal tunnel also play an active role in the regulation of protein synthesis and co-translational folding [485,486]. Ribosomes also perceive each other through quality sensors of collided ribosomes in eukaryotes [487]. In addition, the ribosomes synchronise many complex movements during the translation cycles [488,489,490]. The recent discoveries of “ribosome heterogeneity” [491] also significantly expand the complexity of the possible ribosome’s network topologies [492] and open new perspectives on “network plasticity” that could also play a role in its behavioural richness.

A recent interdisciplinary study has shown how r-protein networks have evolved toward a growing complexity through the coevolution of the r-protein extensions and the increasing number of connexions [46]. This revealed that network expansion is produced by the collective (co)-evolution of r-proteins leading to an asymmetrical evolution of the two subunits. Furthermore, graph theory showed that the network evolution did not occur at random: each new occurring extension and connection gradually relates functional modules and places the functional centres in central positions of the network. The strong selective pressure that is also expressed at the amino acid acquisition links the network architectures and the r-protein phylogeny, thus suggesting that the networks have gradually evolved to sophisticated allosteric pathways. The congruence between independent evolutionary traits indicates that the network architectures evolved to relate and optimise the information spread between functional modules (Figure 1). This network archaeology study has also revealed the existence of a universal network that consists of 49 strictly conserved connections that were probably present before the radiation of the bacteria and archaea [493] (Figure 5).

Interestingly, this primordial network is much more developed in the small ribosomal subunit suggesting that the large subunit network complexity developed in later evolutionary stages. These findings, therefore, suggest that LUCA already possessed such type of molecular networks, with long wires and tiny interfaces. Interestingly, these networks also mix the π-systems of rRNA and aromatic amino acids of proteins for forming conserved structural motifs probably involved in signal transduction. It is, therefore, possible that this ancestral mode of communication has then not only evolved in modern ribosomes but in other macromolecular systems for information transfer and processing. The ribosome opens a window on the first information processing networks, which appeared at the origin of life. They probably diverged towards other cell systems that have been compared to brains, such as the multiple nano-brains described in Baluska’s article published in this special issue [40].

## 5. Evolution of Informational Systems across Scales

### 5.1. Comparison of Signalling Systems

Just as the observation of similar behaviours of cells, plants and organisms with nervous systems, it does not seem unreasonable to imagine that the signalling mechanisms that underlie them share some common points. In this regard and because of their mechanistic and functional analogies with nervous systems, plant roots [53,54], bacterial chemotaxis [13,48,51,52] and ribosomal signalling networks [44,45] have been compared to brains. This analogy assumes both a spatial and temporal scale invariance between molecular and neuron networks: while ribosomal signalling networks, which are no larger than a few tens of nanometres, can tell us about the first life informational systems at the origin of life, nervous systems, of the order of a metre in size, corresponding to a major evolutionary transition that occurred 600 million years ago, the appearance of metazoans [59,115,494,495]. The comparison of these systems and their behaviours reveal similarities and differences that we believe can provide interesting insights into the invariants of living information systems and the way they have evolved during major evolutionary transitions (Figure 6).

The perception, transfer and integration of signals by the ribosomes or the cells ensure autonomous behaviour and appropriate decision making in the face of their own fluctuations or those of their environment. However, if for cells the “outside” consists of other cells, the intercellular space or their ecosystem environment, the ribosomes listen to each other, organelles, and they monitor what may be happening in the cytoplasm. Whether anchored in ribosomal rRNA or membrane phospholipids, the sensor proteins of both systems have developed close relationships with their matrix, which is actively involved in the signalling processes through various mechanisms. While rRNA allostery also participates in information transfer within the ribosome [479], phospholipids modulate the responses of the receptors and may also constitute signalling molecules [285].

However, the main difference between ribosomal and cellular signalling lies in the duration of the interaction between the nodes. Once the ribosome is assembled, interactions between ribosomal proteins and functional centres are mostly “permanent” in that most of the r-protein remain connected (Figure 3 and Figure 4). However, the r-protein nodes can also interact transiently with tRNAs, mRNAs, various translation factors and proteins outside the ribosome. In contrast, the cell signalling networks are essentially transient and involve contacts between proteins that diffuse into the cytoplasm. Note, however, that proteins anchored in membrane systems can also establish long-lasting contacts with their partners in signalling clusters or scaffolds (as, for example, the chemoreceptor and CheA and CheW) (Figure 3). However, cells also display systems analogous to ribosomal extensions involved in signal transmission: the filaments of the cytoskeleton [496]. Thus, the ribosome and the cell signalling networks correspond, respectively, to the solid and liquid brains concepts described by R. Solé [62] (Figure 6).

The structure and duration of the interactions within the network constraints and have a direct implication on the nature of the signals that are transmitted between partners. While the ribosomal “permanent” networks can transmit diverse transient or ephemeral signals that can be grouped into allosteric phenomena, transient networks have had to develop chemical messengers that are durable over time. The long-lasting but reversible post-translational modifications of proteins and the synthesis of second messengers that trigger downstream signalling cascades are signalling solutions to the “impermanence” of networks. An evolutionary consequence of the network transient interactions is a great diversity in the protein modification and signalling codes. Conversely, the evolution of the nervous systems has convergently reinstated networks based on durable interactions between neurons through the establishment of “true” synapses. In consequence, simar to ribosome networks, a unification of signal transmission modes seems to have re-emerged later in evolution: the directional propagation of “ephemeral signals”, the action potentials induced by the discharge of neurotransmitters in the inter-synaptic space. The analogy between the ribosomal network and the nervous system can also be continued with regard to the organisation of functional classes of nodes. For example, the number of nodes in the eukaryotic r-protein network is of the same order of magnitude as the *C. elegans* neural network, which contains several hundred neurons [497]. These two networks are organised into modules dedicated to distinct functions in processing information (Figure 7). Further comparisons between the architectures and modular organisation of networks containing an equivalent number of nodes at different size scales could provide valuable information on the general principles of information processing. However, beyond the analogies concerning the connectivity of networks, data on their dynamics and the way in which signals are exchanged are indispensable.

### 5.2. Bayesian Brain Hypothesis

Graph theory and computational approaches have provided various models for understanding how signalling networks are organised and identified a number of emergent properties such as noise reduction, signal amplification [41,42,498,499,500] or even associative learning [501,502]. In the framework of the Bayesian brain, adaptative systems evolve in such a way that they can predict the behaviour of their environment so that they can preserve their integrity. These “brains” make a hypothesis on the evolution of their environment in order to make sense of it and act accordingly to these hypotheses in a way that maximises their benefits. These systems update, in a Bayesian fashion, the hypothesis they make on the world with respect to new observations, which, in turn, changes what they expect the future outcomes of the environment would be. For example, Lee and Mumford proposed a Bayesian interpretation, using hierachical Bayesian models, of how the visual cortex integrates information accounting for neurophysiological evidence previous paradigm could not account for [503].

A model of the Bayesian brain hypothesis is the active inference framework [504,505], where a “brain” receives information from gateways such as sensory organs and transfers it as information onto its internal model of the world in an optimal fashion. It then uses this information to act in adequacy with the hypothesis is made on how the external world evolves accordingly to its observations.

More precisely, observations of the world, received, for example by sensory organs, are seen as being generated by an underlying unobserved process that the brain can only model partially. In this framework, the “brain” specifies internally collections of variables that stand for the causes on which depend the observations, but as this modelisation is imperfect, the inputs depend on the causes in a stochastic manner. This “brain” uses its observations to choose from different hypotheses on the causes that would have generated this input with respect to its own modelisation of the generative process; this is the inference step, a hypothesis is a probability distribution over the causes, and the “brain” proceeds through this step by choosing the best approximation on the posterior on the causes after new input data are observed. The “brain” then makes use of this hypothesis to act, for example, in such a way as to maximise a reward. There are several propositions for how Bayesian computations could be implemented biologically, for example, in the cortex [504,505,506] in a manner consistent with experimental data on connectivity (role of forward and backward connections) and temporality of cortical response [505,507].

We want to stress that what is important is how biological signalling interact with one another if one wants to understand what “computations” underly such process: biologically relevant quantities act as a dictionary complex enough for encoding important variables over which to take decisions and interactions between such quantities have enough structure to capture the interactions between these variables. For example, in [508], classical equations describing the dynamics of rod phototransduction can be interpreted as a belief update algorithm on the presence or absence of a photon reaching the rod. This point of view motivates the relevance of the use of a Bayesian brain framework to other intelligent systems capable of decision making but lacking a nervous system [509]. More generally, cell signalling can, remarkably, be interpreted in a Bayesian fashion, where macromolecules and messengers are the physical support for computing Bayesian a posterioris [74]. For example, it has been observed that the unicellular *Physarum polycephalum*’s choices when facing two environments with different amounts of reward is dependent in a probabilistic manner on the proportion of reward in each environment; however, information on how signalling occurs is yet to be understood [510].

### 5.3. Beyond Allostery? Aromaticity in Signal Integration and Decision Making

From a mechanistic point of view, the central question is, how do molecular or brain networks integrate multiple signals and take appropriate decisions? For example, in ribosomal networks, many r-proteins display multiple connections where some hubs such as the eukaryotic uL4 or uS8 can be connected to more than eight partners [45,46] (Figure 1). What are the molecular mechanisms underlying the integration and decision making from these multiple stimuli? The secret is in the modular organisation and the combinatory interaction of modules at different scale levels.

At the cellular level, the combinatorial binding of multiple receptors and adaptators allows signal integration in space and time. For example, RTKs can recognise specific targets and integrate multiple stimuli through regulated multidomain interactions. Thus, multiple signal inputs can be integrated through a combination of recognition modules and permit “coincidence” detection [347,349,511]. At the molecular level, the detection and integration of multiple signals rely on sophisticated allosteric mechanisms observed in particular proteins. Similarly, we will see that it is the combinatorial association of smaller “allosteric modules” within protein domains that make possible the integration of multiple signals.

However, allostery, which, according to Jacques Monod, constitutes “the second secret of life” [512], is an evolving concept. A classic view is that allostery consists of a remote “communication” between two sites of a macromolecule, an “allosteric” or regulatory site and a “functional” or catalytic site. Thus a ligand that binds at a distance from an enzyme active site may modulate (decreasing or increasing) its activity. However, the conceptualisation of “how signals are propagated” along a macromolecule to regulate distant sites has evolved considerably. Allostery has transited from the simple concept where conformational changes were propagated in a “domino-like” motion to a more probabilistic view based on the multiple dynamic states of proteins and “conformation ensembles” [286,513]. According to Nussivov and colleagues, a unified view bringing together a thermodynamic and structural approach now provides a better understanding of how allostery works [121]. Today, we more readily speak of structural coupling between functional and allosteric sites and of “allostetric propagation pathway”, which establishes can form “channels” or “networks” within or between macromolecules (RNA or proteins). Coupling the two distinct sites would mean creating a concerted or correlated movement between them in the dynamics of the protein through a network of interaction between the residues between these two sites. There is now a large repertoire of mechanisms that macromolecules can use to transmit intra- or inter-molecular signals, and in recent decades, the molecular disorder has also contributed to documenting allosteric mechanisms. It is thought that propagations of order-disorder transitions can contribute to signal transmission [514]. In addition, intrinsically unstructured proteins are thought to play a particular role in signal amplification due to the low energy barriers between conformers [515,516,517,518].

Allostery is itself subject to evolutionary processes throughout the history of life. In the same way as sensitivity to physico-chemical stimuli and due to their intrinsic dynamics, allostery is a ubiquitous property of biological macromolecules [519,520,521,522]. Evolution has been able to modulate allosteric properties along particular pathways by optimising the sequences of macromolecules [523]. On the other hand, allosteric regulation and catalysis have emerged from common pathways since they share evolutionary optimisation of the same conformational mobility in protein sequence [524,525]. Evolution has thus selected robust sequences capable of compensating for the deleterious effects of mutations in allosteric pathways [526]. It is now widely accepted that macromolecules have optimised their sequences during evolution to inhabit, represent (“populate”) several states that can switch from one to another (switchable states), e.g., one disordered or unfolded and one ordered.

However, macromolecules also display other signal transmission mechanisms involving their electrostatic properties. For example, the coupling of structural changes between distant domains can be electrostatically driven and rely on the reorganisation of electrostatic charges and potentials [527,528]. For example, the charge reorganisation mediates the communication between the two domains of the bilobed r-protein bL20 [527] and calmodulin [529]. These mechanisms have subsequently been observed in other systems [530] and are referred to as dielectric allostery [531,532]. Dipole coupling and electrodynamic processes may also be involved in signal propagation between distant sites in proteins [533,534].

Well-documented structural studies have provided valuable insights into how either enzymes or receptors have developed sophisticated mechanisms to integrate multiple signals from remote sites: multisite allosteric enzymes and a dual-sensor histidine kinase. These studies help to decipher the general principles of signal integration at the molecular levels and provide structural and dynamical insights that may contribute to elucidating the intertwined molecular mechanisms of information transfer and processing in “molecular brains”. Enzymatic multisite allostery and receptor allosteric modulation share similar features that are described here. Multisite allostery or synergistic allostery allows enzymes to integrate and modulate their catalytic properties according to the presence of several substrates. A well-characterised example is the 3-deoxy-D-*arabino* heptulosonate 7-phosphate synthase (DAH7PS) of *Mycobacterium tuberculosis*, the first enzyme of the shikimate pathway, a metabolic pathway responsible for the aromatic amino acids [535]. This tetrameric enzyme whose monomer adopts a TIM-barrel fold possess distinct and remote allosteric sites for each aromatic Trp, Phe and Tyr amino acids. Their individual or combinatory binding affects differently the catalytic activity [536,537,538,539]. Interestingly, the structures of the free and ligand-bound enzymes do not display significant structural change suggesting that the allostery proceeds with the modification of the backbone dynamics. These works performed in Emily Parker’s lab are very valuable because it allows us to follow in detail the structural mechanisms at the origin of the integration of these multiple signals.

On the other hand, many receptors are allosterically are regulated by two inter-related sites. The activation of the orthosteric site by the orthosteric ligand is modulated by the allosteric site, which binds the allosteric modulators [286,307]. The binding of the ligands on the two remote sites modulates the receptor activation and the downstream cell signalling cascade. A recent dynamic crystallographic study on a dual-sensor histidine kinase has deciphered the allosteric mechanisms underlying the integration of its perception of light and phosphorylation signals. This system that combines a set of sensory and allosteric modules operates a molecular logic OR [540]. Interestingly these multi-sensor proteins share many structural and dynamical features found in r-protein networks. First, they combine diverse interacting sensor and allosteric modules that use different mechanisms for sensing and propagating signals. Second, these modules that have probably co-evolved for integrating multiple signals have acquired highly conserved aromatic residues that play critical and perhaps still not well-understood roles in both information transfer and processing [46]. For example, particular motifs formed between aromatic and charged residues (such as cation-π) mediate the communication between the allosteric modules or form the “ligand-binding pocket” in most of the membrane receptors [310,541] (Figure 8).

Third, in both the study of r-protein networks and receptors, many experimental observations do not always support a classical view of allostery. For example, allosteric ON/OFF switches often cannot be distinguished by marked structural differences in many receptors in different ribosomal functional states [45]. Furthermore, loose dynamical coupling is observed between domains that are supposed to cooperate [17]. Together, these observations suggest other types of allosteric mechanisms in which aromatic amino acids are involved in the propagation of electrostatic perturbations or charge transfer through their π electrons [45], as in nanowires [542] or DNA [543]. Thus, the structural motifs formed by the combination of aromatics and charged amino acids could give rise to subtle modulations of the signals that transit along proteins and play a critical role in both the transfer and integration of these signals. Could the structural analogy between neural networks and protein networks also be continued on a dynamic level? Could there be, for example, like the “neural code” based on the information contained in spike trains, an “allosteric code” at the basis of information processing in molecular networks? This question opens new perspectives on allosteric mechanisms and could initiate the exploration of a new paradigm.

And to close the loop, the cryptochromes of monarch butterflies bring quantum mechanics into biology thanks to their tryptophan clusters and their π-electrons [8,9]. Thus, aromatic amino acids have properties that are still underestimated, and in the light of their critical and still poorly understood role in allosteric processes, it does not seem unreasonable to us to imagine that they contribute to the processing of information through quantum phenomena [65]. These findings may provide useful insights for designing organic processors for the computers of the future.

## 6. Conclusions

From its earliest forms of organisation, life has had to accommodate and adapt to incessant fluctuations coming from both its internal activity and the external environment. Thus, in addition to its properties of self-organisation [544,545] and self-replication, which are considered its essence, life has had to develop complex behaviours from its origins, and its “first wills” already needed a “molecular brain”. Perceiving and integrating various signals and responding to them by taking appropriate decisions to survive and eventually prosper is, therefore, an indispensable faculty for all forms of life, regardless of its degree of organisation. Thus, when considering the “emergence of consciousness” from a molecular network, the two questions “what is the soul (the *psychès* of Aristotle) and what is life” may converge into one and the same and unique question.

## Figures and Tables

**Figure 1 ijms-22-11868-f001:**
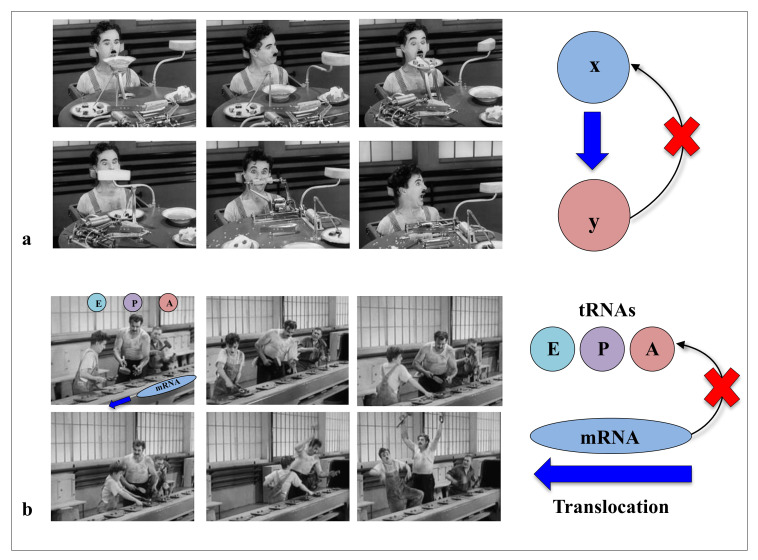
“Feeding without feedback”. In his movie “Modern Times”, Chaplin illustrates the limits of the machine metaphors in biology. (**a**) A “machine” is supposed to feed the worker without feed-back. (**b**) The scene showing three workers who, in order to screw in bolts, have to follow the rapid and uncontrollable movement of a conveyor belt illustrates in a suggestive way the difficult task of the three tRNAs (A, P and E) in a "machine" ribosome without any feedback process between the different actors.

**Figure 2 ijms-22-11868-f002:**
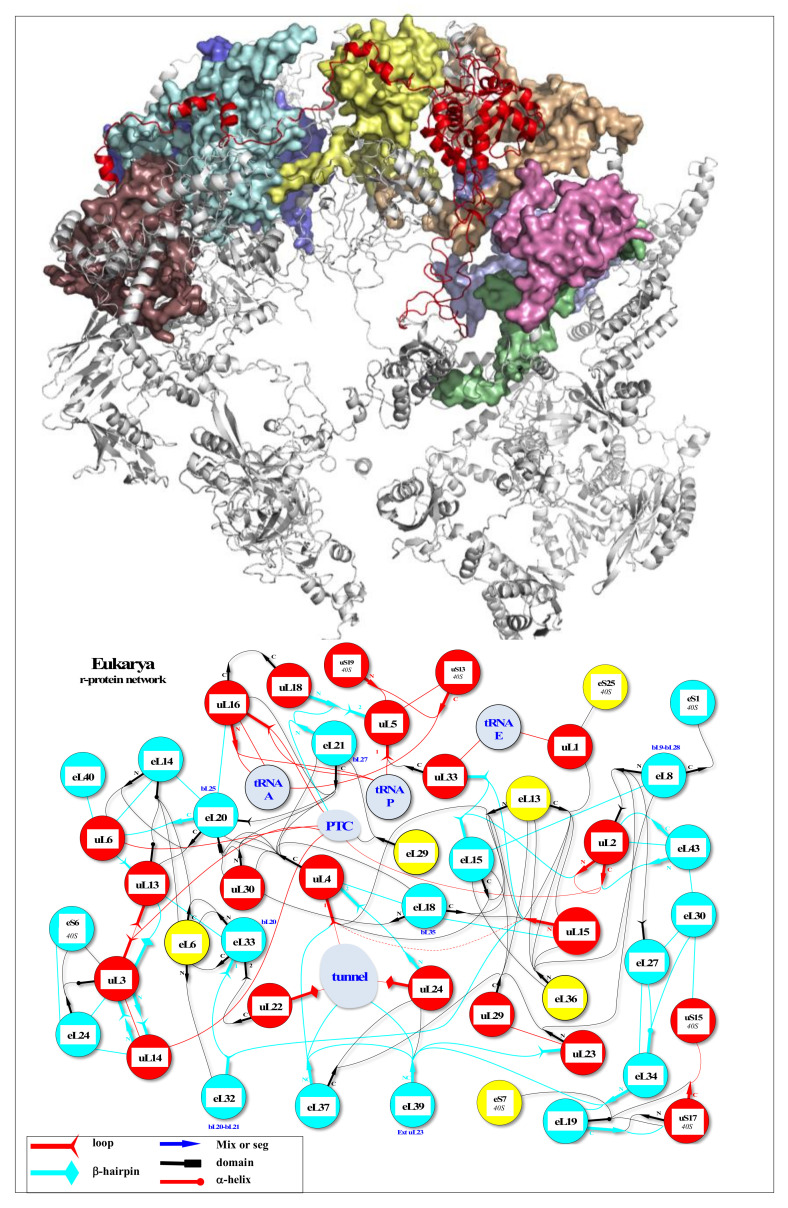
r-protein networks in the large subunit of the eukaryotic ribosome. (Top) Cartoon representation of the large subunit of the eukaryotic ribosome (pdb_id: 4v88) [55]. The protein hub uL4 is represented in red, and its numerous partners are represented by coloured surfaces. (Bottom) Evolution of the r-protein network by accretion around a universal network (red proteins): the r-proteins and their extensions are represented according to their evolutionary status. Red: universal (common to bacteria, archaea and eukarya); cyan: archaea; yellow: eukarya. Coloured thin lines symbolise an interaction between the r-protein extensions (the legend of the line codes for the extensions is indicated in the box). Lines between two circles symbolise an interaction between two globular domains. The colours of the lines follow the code for the evolutionary status described above, except for eukarya-specific connections that are represented with black lines for clarity. “N” or “C” indicate if the seg or mix are N-terminal or C-terminal extensions. NC indicates proteins without a globular domain (uS14, eL29, eS30, eL37 and eL39). Functional sites (PTC, Tunnel, tRNAs and mRNA) are represented in light blue. The names of bacterial proteins, which, by convergence, occupy a position similar to that of eukaryotic or archaeal r-proteins are shown in blue below the circles.

**Figure 3 ijms-22-11868-f003:**
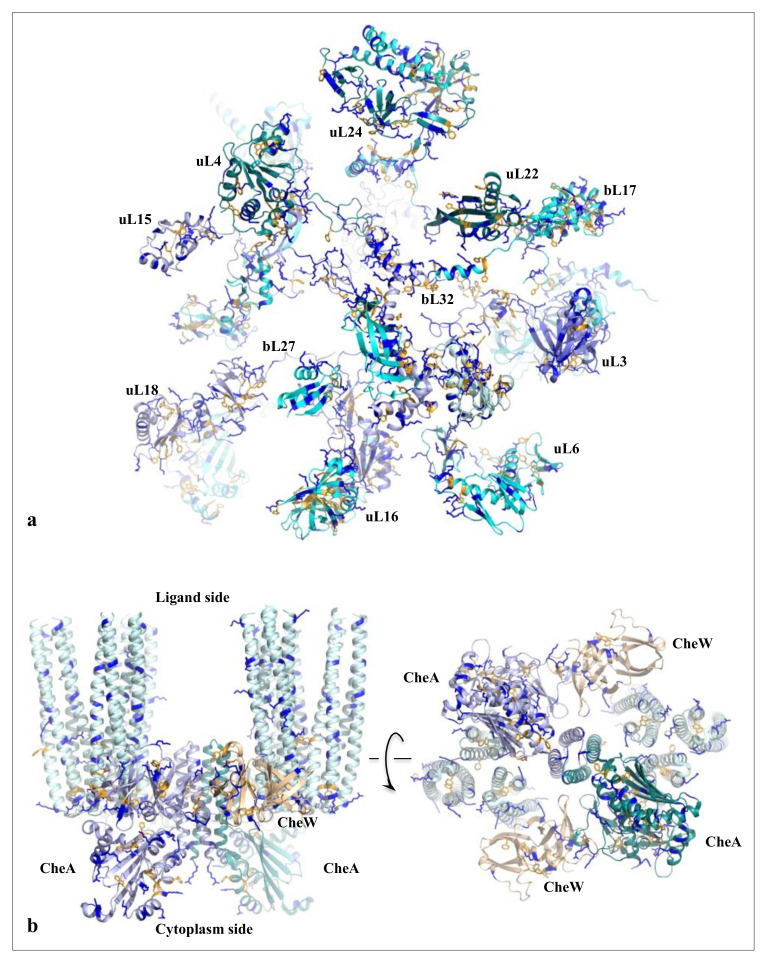
Structural comparison of two molecular brains. (**a**) The r-protein network of the bacterial ribosome (pdb_id 4y4p). For clarity, the rRNA is not represented. (**b**) The structure of a chemosensory array involved in the bacterial chemotaxis (pdb_id 6s1k). Both structures are shown at the same scale.

**Figure 4 ijms-22-11868-f004:**
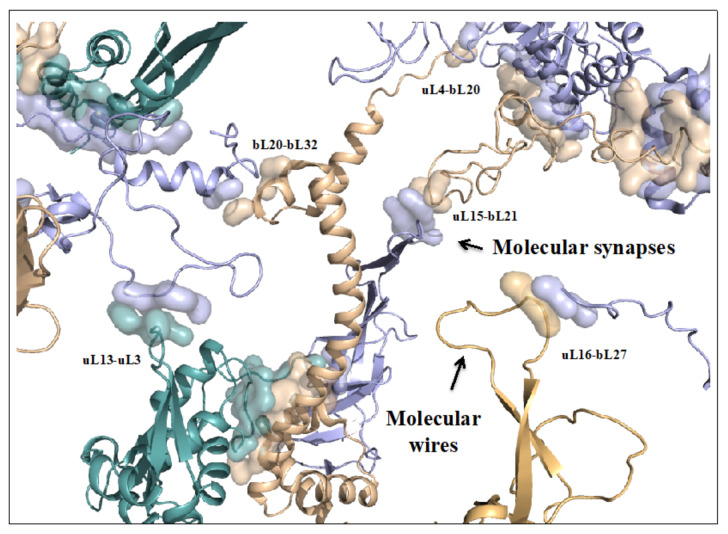
Molecular synapses and wires in the bacterial large subunit r-protein network. The tiny interfaces (the molecular synapses) between r-proteins are represented by surfaces. rRNA is not represented for clarity.

**Figure 5 ijms-22-11868-f005:**
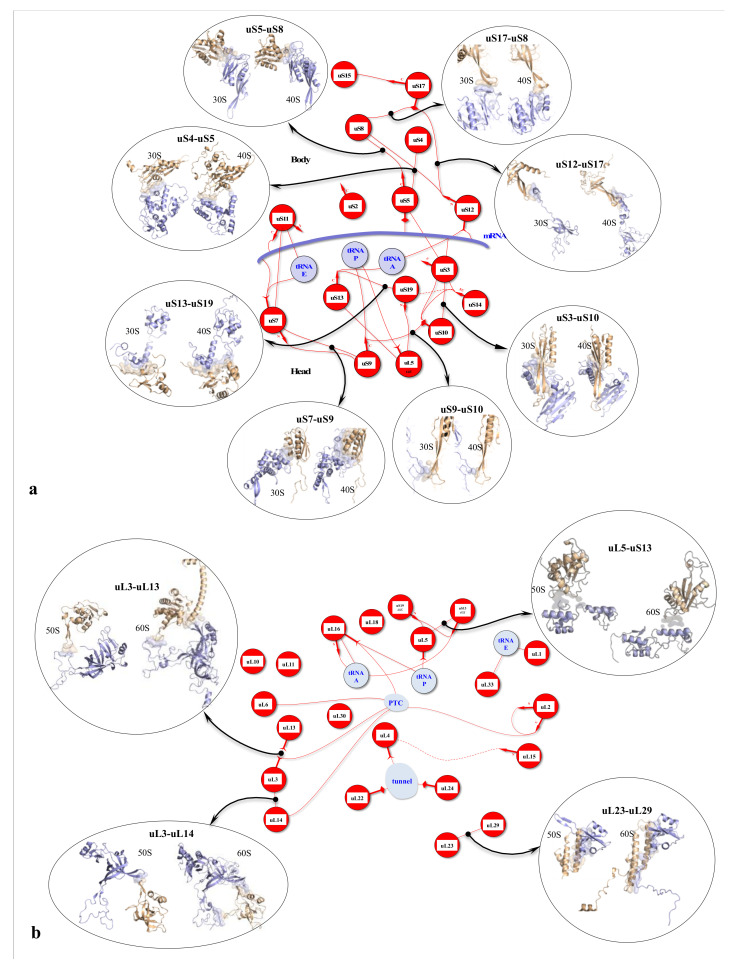
Universal r-protein network that probably predated the radiation of the three kingdoms. The interaction common in the small (**a**) and large (**b**) subunits of the ribosomal structures is shown by blue cartoons [46]. Conserved contacts between bacterial and eukaryotic ribosomal proteins are indicated by lines between the nodes of the network and represented by cartoons in the corresponding discs (left: bacteria (30S or 50S); right: eukaryotes (40S or 60S)).

**Figure 6 ijms-22-11868-f006:**
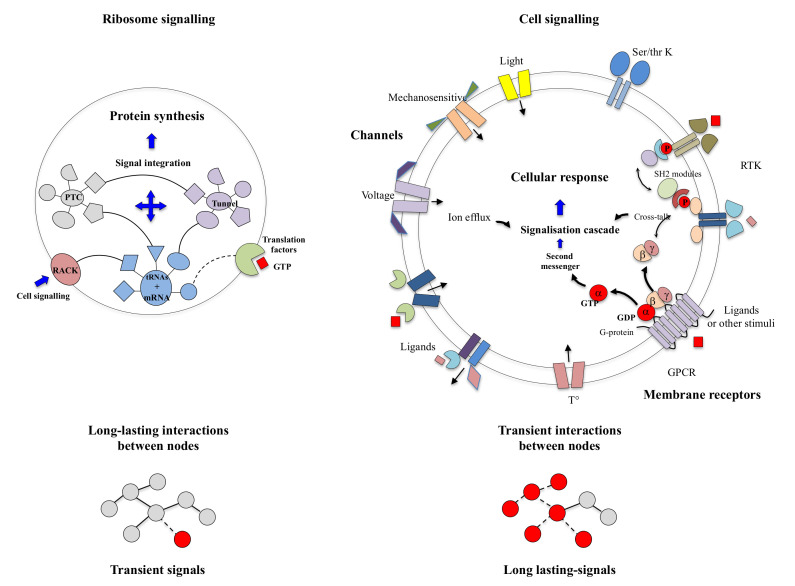
Comparison of ribosome and cell signalling. (**Left**) Long-lasting interactions between network nodes (ribosomal proteins and functional centres) in the ribosome are suitable for the propagation of ephemeral signals. (**Right**) Transient interactions between nodes in cell signalling impose the development of long-lasting diffusible signals (post-translational modification or synthesis of second messengers).

**Figure 7 ijms-22-11868-f007:**
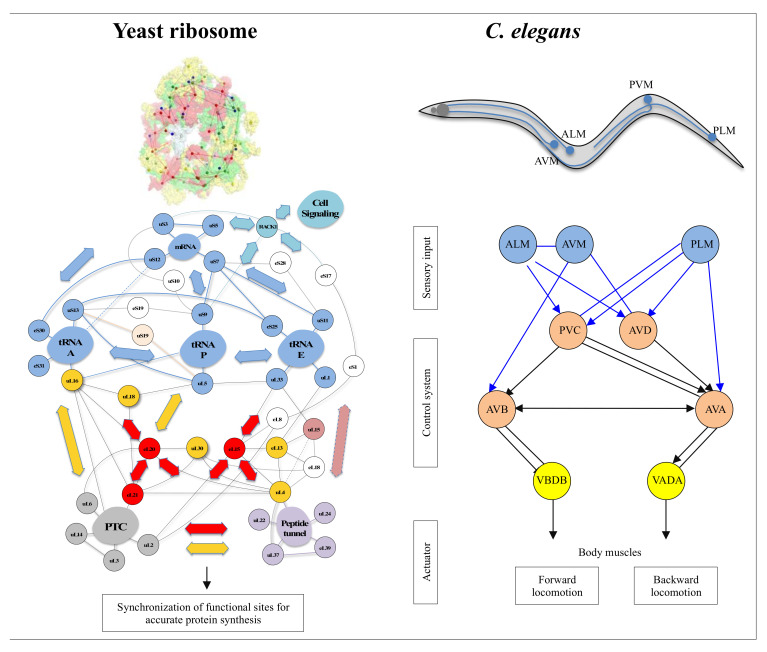
Functional analogy and modular organisation of the nodes in a ribosomal network (eukaryotic) and in a subnetwork of the nervous system of a simple organism (*C. elegans*, adapted from ref [497]).

**Figure 8 ijms-22-11868-f008:**
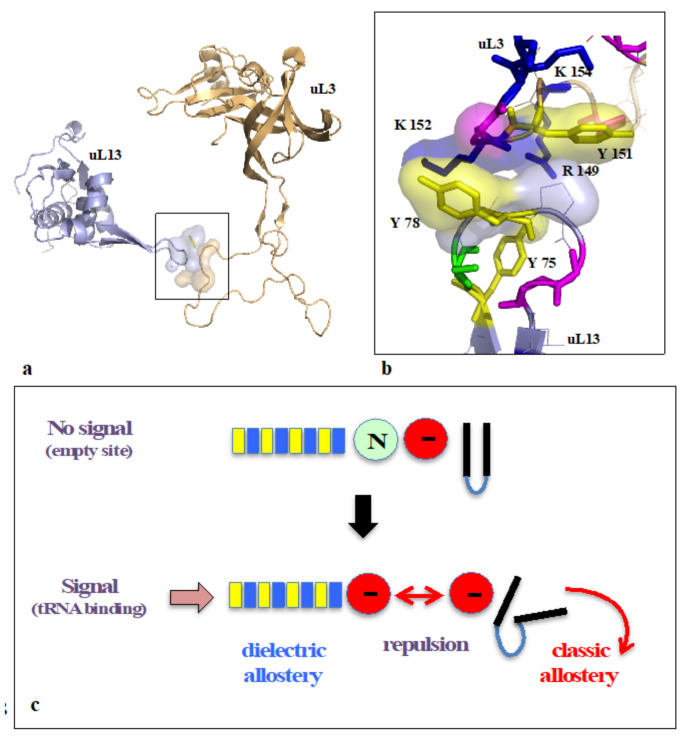
Aromatic amino acids in information transfer and processing. (**a**,**b**): global and detailed view of a “molecular synapse” in bacterial large subunit r-protein network. The conserved amino acids are represented by coloured sticks. (**c**) Possible pathways of electrostatic signalling through an array of action-π interactions (yellow rectangles = aromatic residues and blue rectangles are basic residues). The combination of diverse allosteric modules can integrate multiple stimuli. The transient change of charge of an amino acid such as a histidine induced by its electrostatic context may induce large conformational changes (“N” = neutral and “−” = negative charge).

## Data Availability

Not applicable.

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
