# Peer review of "Towards the Idea of Molecular Brains"

_ijms, 2021, doi:10.3390/ijms222111868_

Round 1
Reviewer 1 Report
I have enjoyed reading this very stimulating manuscript. It opens doors to a new science of molecular nanobrains which will improve our understanding of cells very much. This is absolutely essential for our further advances in still elusive understanding of life as such, but also in nanotechnologies relevant for medicine in future. Authors may consider to add into their discussion the following paper:
Nano-intentionality: a defense of intrinsic intentionality. Biology & Philosophy 23: 157-177 (2008)
Author Response
We are very grateful for the encouragement of reviewer 1. These comments encourage us to continue our work in this direction.
We also thank reviewer 1 for pointing out the very interesting article on nanointentionality that we quote from the introduction on page 3
Reviewer 2 Report
Well, that is definitely a very interesting perspective – beautifully narrated, “in a poetic way”!
Some minor typos detected (e.g.: `brain').
What can I possibly say more?!?!
Undoubtedly, only artists devote themselves to science . . .
Undoubtedly!
Author Response
we sincerely thank reviewer 2 for his encouraging comments which stimulate us to continue our research in this direction
Reviewer 3 Report
October 6, 2021
Review of the paper (ijms-1410537-peer-review-v1)
Towards the idea of molecular brains
by Youri Timsit, and Gregoire Sergeant-Perthuis, submitted to the section: Biochemistry,
https://www.mdpi.com/journal/ijms/sections/biochemistry
>From Nanomachine to Nanobrain, Information Processing at a Molecular Scale
https://www.mdpi.com/journal/ijms/special_issues/nano-machines
General comments
In this paper, the Author aim to review the evolution of the information processing system taking into account their scales in the context of designing organic computers. I find the idea presented in the paper very interesting and inspiring but I have the following comments and concerns:
Problems missed (a reference to real transmission mechanisms)
When reading this interesting and inspiring article, there seems to be a visible lack of a reference to the way that neurons communicate (neural codes). We can not discuss the brain without pointing out this important issue. Since Adrian and Zotterman papers it has been established that in the nervous system information is transmitted among spiking neurons by trains of discrete electrical pulses, called action potentials or spikes. The challenges and analysis in this area are summarized in the books by:
J. Leo van Hemmen, T.J. Sejnowski "23 Problems in Systems Neuroscience" Oxford University Press, 2006.
Gerstner W, Kistler WM, Naud R, Paninski L (2014) Neuronal dynamics. “From single neurons to networks and models of cognition.” Cambridge University Press, Cambridge.
On the other hand, the quantitative mathematical approach to the concept of information, which is described in the book by Cover, T.M .; Thomas, J.A., Elements of Information Theory; Wiley: New York, NY, USA, 1991 should be mentioned as Authors indicate computer building as the area of application. Also, interesting analysis concerning information processing addressing “neural coding” challenges were developed in the papers:
Pregowska, A.; Kaplan, E.; Szczepanski, J., How Far can Neural Correlations Reduce Uncertainty? Comparison of Information Transmission Rates for Markov and Bernoulli Processes. Int. J. Neural. Syst. (2019), 29 (8).
R. Brette, Philosophy of the spike: Rate-based versus spike-based theories of the brain, Front. Syst. Neurosci. Rev. 9 (2015) 151.
Crumiller M, Knight B, Kaplan E (2013), The measurement of information transmitted by a neural population: promises and challenges. Entropy 15(9):3507–3527.
P. M. Di Lorenzo, J. Y. Chen and J. D. Victor, Quality time: Representation of a multidimensional sensory domain through temporal coding. J. Neurosci. 29 (2009) 9227–9238.
I recommend including into the proposed review paper a Section, or at least a Paragraph, dealing with these important, in particular taking into application field, aspects of information processing by neural systems among others on the above references.
Final comments
The idea presented and developed in the paper is promising, interesting, and important, but due to the above concerns, I could recommend this paper for publication provided that the above comments/problems will be carefully addressed. At this moment I would recommend Major Revision.
Author Response
we thank reviewer 3 for his valuable advice. We fully agree with his remarks: a paragraph on the question of the nature of the signals exchanged in the brains and the neural code was indeed missing. We have introduced this paragraph on page 8, at the end of section 1.4 "brain beyond connectomes" and quoted all the references that have been proposed
Round 2
Reviewer 3 Report
The authors took my comments into account and recommends the paper for publication.